# Differentiable Trust Region Layers for Deep Reinforcement Learning

**Fabian Otto**[*]
Bosch Center for Artificial Intelligence
University of Tübingen

**Philipp Becker**
Karlsruhe Institute of Technology

**Ngo Anh Vien & Hanna Carolin Ziesche**
Bosch Center for Artificial Intelligence

**Gerhard Neumann**
Karlsruhe Institute of Technology

## Abstract

Trust region methods are a popular tool in reinforcement learning as they yield robust policy updates in continuous and discrete action spaces. However, enforcing such trust regions in deep reinforcement learning is difficult. Hence, many approaches, such as *Trust Region Policy Optimization* (TRPO) and *Proximal Policy Optimization* (PPO), are based on approximations. Due to those approximations, they violate the constraints or fail to find the optimal solution within the trust region. Moreover, they are difficult to implement, often lack sufficient exploration, and have been shown to depend on seemingly unrelated implementation choices. In this work, we propose differentiable neural network layers to enforce trust regions for deep Gaussian policies via closed-form projections. Unlike existing methods, those layers formalize trust regions for each state individually and can complement existing reinforcement learning algorithms. We derive trust region projections based on the Kullback-Leibler divergence, the Wasserstein L2 distance, and the Frobenius norm for Gaussian distributions. We empirically demonstrate that those projection layers achieve similar or better results than existing methods while being almost agnostic to specific implementation choices. The code is available at `https://git.io/Jthb0`.

## 1 Introduction

Deep reinforcement learning has shown considerable advances in recent years with prominent application areas such as games (Mnih et al., 2015; Silver et al., 2017), robotics (Levine et al., 2015), and control (Duan et al., 2016). In policy search, *policy gradient* (PG) methods have been highly successful and have gained, among others, great popularity (Peters & Schaal, 2008). However, often it is difficult to tune learning rates for vanilla PG methods, because they tend to reduce the entropy of the policy too quickly. This results in a lack of exploration and, as a consequence, in premature or slow convergence. A common practice to mitigate these limitations is to impose a constraint on the allowed change between two successive policies. Kakade & Langford (2002) provided a theoretical justification for this in the approximate policy iteration setting. Two of the arguably most favored policy search algorithms, *Trust Region Policy Optimization* (TRPO) (Schulman et al., 2015a) and *Proximal Policy Optimization* (PPO) (Schulman et al., 2017), follow this idea using the *Kullback-Leibler divergence* (KL) between successive policies as a constraint.

We propose closed-form projections for Gaussian policies, realized as differentiable neural network layers. These layers constrain the change in successive policies by projecting the updated policy onto trust regions. First, this approach is more stable with respect to what Engstrom et al. (2020) refer to as *code-level optimizations* than other approaches. Second, it comes with the benefit of imposing constraints for individual states, allowing for the possibility of state-dependent trust regions. This allows us to constrain the state-wise maximum change of successive policies. In this we differ from previous works, that constrain only the expected change and thus cannot rely on exact guarantees of monotonic improvement. Furthermore, we propose three different similarity measures, the KL

---

[*]Correspondence to fabian.otto@bosch.com

divergence, the Wasserstein L2 distance, and the Frobenius norm, to base our trust region approach on. The last layer of the projected policy is now the the trust region layer which relies on the old policy as input. This would result in a ever-growing stack of policies, rendering this approach clearly infeasible. To circumvent this issue we introduce a penalty term into the reinforcement learning objective to ensure the input and output of the projection stay close together. While this still results in an approximation of the trust region update, we show that the trust regions are properly enforced. We also extend our approach to allow for a controlled evolution of the entropy of the policy, which has been shown to increase the performance in difficult exploration problems (Pajarinen et al., 2019; Akrour et al., 2019).

We compare and discuss the effect of the different similarity measures as well as the entropy control on the optimization process. Additionally, we benchmark our algorithm against existing methods and demonstrate that we achieve similar or better performance.

## 2    RELATED WORK

**Approximate Trust Regions.**    Bounding the size of the policy update in policy search is a common approach. While Kakade & Langford (2002) originally focused on a method based on mixing policies, nowadays most approaches use KL trust regions to bound the updates. Peters et al. (2010) proposed a first approach to such trust regions by formulating the problem as a constraint optimization and provided a solution based on the dual of that optimization problem. Still, this approach is not straightforwardly extendable to highly non-linear policies, such as neural networks. In an attempt to transfer those ideas to deep learning, TRPO (Schulman et al., 2015a) approximates the KL constraint using the Fisher information matrix and natural policy gradient updates (Peters & Schaal, 2008; Kakade, 2001), along with a backtracking line search to enforce a hard KL constraint. Yet, the resulting algorithm scales poorly. Thus, Schulman et al. (2017) introduced PPO, which does not directly enforce the KL trust region, but clips the probability ratio in the importance sampling objective. This allows using efficient first-order optimization methods while maintaining robust training. However, Engstrom et al. (2020) and Andrychowicz et al. (2020) recently showed that implementation choices are essential for achieving state-of-the-art results with PPO. *Code-level optimizations*, such as reward scaling as well as value function, observation, reward, and gradient clipping, can even compensate for removing core parts of the algorithm, e. g. the clipping of the probability ratio. Additionally, PPO heavily relies on its exploration behavior and might get stuck in local optima (Wang et al., 2019). Tangkaratt et al. (2018) use a closed-form solution for the constraint optimization based on the method of Lagrangian multipliers. They, however, require a quadratic parametrization of the Q-Function, which can limit the performance. Pajarinen et al. (2019) introduced an approach based on compatible value function approximations to realize KL trust regions. Based on the reinforcement learning as inference paradigm (Levine, 2018), Abdolmaleki et al. (2018) introduced an actor-critic approach using an Expectation-Maximization based optimization with KL trust regions in both the E-step and M-step. Song et al. (2020) proposed an on-policy version of this approach using a similar optimization scheme and constraints.

**Projections for Trust Regions.**    Akrour et al. (2019) proposed *Projected Approximate Policy Iteration* (PAPI), a projection-based solution to implement KL trust regions. Their method projects an intermediate policy, that already satisfies the trust region constraint, onto the constraint bounds. This maximizes the size of the update step. However, PAPI relies on other trust region methods to generate this intermediary policy and cannot operate in a stand-alone setting. Additionally, the projection is not directly part of the policy optimization but applied afterwards, which can result in sub-optimal policies. In context of computational complexity, both, TRPO and PAPI, simplify the constraint by leveraging the expected KL divergence. Opposed to that, we implement the projections as fully differentiable network layers and directly include them in the optimization process. Additionally, our projections enforce the constraints per state. This allows for better control of the change between subsequent policies and for state-dependent trust regions.

For the KL-based projection layer we need to resort to numerical optimization and implicit gradients for convex optimizations (Amos & Kolter, 2017; Agrawal et al., 2019). Thus, we investigate two alternative projections based on the Wasserstein L2 and Frobenius norm, which allow for closed form solutions. Both, Wasserstein and Frobenius norm, have found only limited applications in reinforcement learning. Pacchiano et al. (2020) use the Wasserstein distance to score behaviors of

agents. Richemond & Maginnis (2017) proposed an alternative algorithm for bandits with Wasserstein based trust regions. Song & Zhao (2020) focus on solving the trust region problem for distributional policies using both KL and Wasserstein based trust regions for discrete action spaces. Our projections are applicable independently of the underlying algorithm and only assume a Gaussian policy, a common assumption for continuous action spaces.

Several authors (Dalal et al., 2018; Chow et al., 2019; Yang et al., 2020) used projections as network layers to enforce limitations in the action or state space given environmental restrictions, such as robotic joint limits.

**Entropy Control.** Abdolmaleki et al. (2015) introduced the idea of explicitly controlling the decrease in entropy during the optimization process, which later was extended to deep reinforcement learning by Pajarinen et al. (2019) and Akrour et al. (2019). They use either an exponential or linear decay of the entropy during policy optimization to control the exploration process and escape local optima. To leverage those benefits, we embed this entropy control mechanism in our differentiable trust region layers.

## 3 PRELIMINARIES AND PROBLEM STATEMENT

We consider the general problem of a policy search in a *Markov Decision Process* (MDP) defined by the tuple $(\mathcal{S}, \mathcal{A}, \mathcal{T}, \mathcal{R}, \mathcal{P}_0, \gamma)$. We assume the state space $\mathcal{S}$ and action space $\mathcal{A}$ are continuous and the transition probabilities $\mathcal{T} : \mathcal{S} \times \mathcal{A} \times \mathcal{S} \to [0, 1]$ describe the probability transitioning to state $s_{t+1} \in \mathcal{S}$ given the current state $s_t \in \mathcal{S}$ and action $a_t \in \mathcal{A}$. We denote the initial state distributions as $\mathcal{P}_0 : \mathcal{S} \to [0, 1]$. The reward returned by the environment is given by a function $\mathcal{R} : \mathcal{S} \times \mathcal{A} \to \mathbb{R}$ and $\gamma \in [0, 1)$ describes the discount factor. Our goal is to maximize the expected accumulated discounted reward $R^\gamma = \mathbb{E}_{\mathcal{T}, \mathcal{P}_0, \pi} \left[ \sum_{t=0}^\infty \gamma^t \mathcal{R}(s_t, a_t) \right]$. To find the optimal policy, traditional PG methods often make use of the likelihood ratio gradient and an importance sampling estimator. Moreover, instead of directly optimizing the returns, it has been shown to be more effective to optimize the advantage function as this results in an unbiased estimator of the gradient with less variance

$$\max_\theta \hat{J}(\pi_\theta, \pi_{\theta_{\text{old}}}) = \max_\theta \mathbb{E}_{(s,a) \sim \pi_{\theta_{\text{old}}}} \left[ \frac{\pi_\theta(a|s)}{\pi_{\theta_{\text{old}}}(a|s)} A^{\pi_{\theta_{\text{old}}}}(s, a) \right], \tag{1}$$

where $A^\pi(s, a) = \mathbb{E}[R^\gamma | s_0 = s, a_0 = a; \pi] - \mathbb{E}[R^\gamma | s_0 = s; \pi]$ describes the advantage function, and the expectation is w.r.t $\pi_{\theta_{\text{old}}}$, i.e. $s' \sim \mathcal{T}(\cdot | s, a), a \sim \pi_{\theta_{\text{old}}}(\cdot | s), s_0 \sim \mathcal{P}_0(s_0), s \sim \rho_{\pi_{\theta_{\text{old}}}}$ where $\rho_{\pi_{\theta_{\text{old}}}}$ is a stationary distribution of policy $\pi_{\theta_{\text{old}}}$. The advantage function is commonly estimated by *generalized advantage estimation* (GAE) (Schulman et al., 2015b). Trust region methods use additional constraints for the given objective. Using a constraint on the maximum KL over the states has been shown to guarantee monotonic improvement of the policy (Schulman et al., 2015a). However, since all current approaches do not use a maximum KL constraint but an expected KL constraint, the guarantee of monotonic improvement does not hold exactly either. We are not aware of such results for the W2 distance or the Frobenius norm.

For our projections we assume Gaussian policies $\pi_{\theta_{\text{old}}}(a_t | s_t) = \mathcal{N}(a_t | \mu_{\text{old}}(s_t), \Sigma_{\text{old}}(s_t))$ and $\pi_\theta(a_t | s_t) = \mathcal{N}(a_t | \mu(s_t), \Sigma(s_t))$ represent the old as well as the current policy, respectively. We explore three trust regions on top of Equation 1 that employ different similarity measures between old and new distributions, more specifically the frequently used reverse KL divergence, the Wasserstein L2 distance, and the Frobenius norm.

**Reverse KL Divergence.** The KL divergence between two Gaussian distributions with means $\mu_1$ and $\mu_2$ and covariances $\Sigma_1$ and $\Sigma_2$ can generally be written as

$$\text{KL}(\{\mu_1, \Sigma_1\} \parallel \{\mu_2, \Sigma_2\}) = \frac{1}{2} \left[ (\mu_2 - \mu_1)^T \Sigma_2^{-1} (\mu_2 - \mu_1) + \log \frac{|\Sigma_2|}{|\Sigma_1|} + \text{tr}\{\Sigma_2^{-1} \Sigma_1\} - d \right],$$

where $d$ is the dimensionality of $\mu_1, \mu_2$. The KL uses the Mahalanobis distance to measure the similarity between the two mean vectors. The difference of the covariances is measured by the difference in shape, i.e., the difference in scale, given by the log ratio of the determinants, plus the difference in rotation, given by the trace term. Given the KL is non-symmetric, it is clearly not a distance, yet still a frequently used divergence between distributions. We will use the more common reverse KL for our trust region, where the first argument is the new policy and the second is the old policy.

**Wasserstein Distance.** The Wasserstein distance is a distance measure based on an optimal transport formulation, for more details see Villani (2008). The Wasserstein-2 distance for two Gaussian distributions can generally be written as

$$\mathcal{W}_2\left(\{\mu_1, \Sigma_1\}, \{\mu_2, \Sigma_2\}\right) = |\mu_1 - \mu_2|^2 + \text{tr}\left(\Sigma_1 + \Sigma_2 - 2\left(\Sigma_2^{1/2}\Sigma_1\Sigma_2^{1/2}\right)^{1/2}\right).$$

A key difference to the KL divergence is that the Wasserstein distance is a symmetric distance measure, i.e., $\mathcal{W}_2(q, p) = \mathcal{W}_2(p, q)$. Our experiments also revealed that it is beneficial to measure the W2 distance in a metric space defined by the covariance of the old policy distribution, denoted here as $\Sigma_2$, as the distance measure is then more sensitive to the data-generating distribution. The W2 distance in this metric space reads

$$\mathcal{W}_{2,\Sigma_2}\left(\{\mu_1, \Sigma_1\}, \{\mu_2, \Sigma_2\}\right) = (\mu_2 - \mu_1)^T\Sigma_2^{-1}(\mu_2 - \mu_1)$$
$$+ \text{tr}\left(\Sigma_2^{-1}\Sigma_1 + \mathbb{I} - 2\Sigma_2^{-1}\left(\Sigma_2^{1/2}\Sigma_1\Sigma_2^{1/2}\right)^{1/2}\right).$$

**Frobenius Norm.** The Frobenius norm is a matrix norm and can directly be applied to the difference of the covariance matrices of the Gaussian distributions. To measure the distance of the mean vectors, we will, similar to the KL divergence, employ the Mahalanobis distance as this empirically leads to an improved performance in comparison to just taking the squared distance. Hence, we will denote the following metric as Frobenius norm between two Gaussian distributions

$$F(\{\mu_1, \Sigma_1\}, \{\mu_2, \Sigma_2\}) = (\mu_2 - \mu_1)^T\Sigma_2^{-1}(\mu_2 - \mu_1) + \text{tr}\left((\Sigma_2 - \Sigma_1)^T(\Sigma_2 - \Sigma_1)\right).$$

The Frobenius norm also constitutes a symmetric distance measure.

## 4 Differentiable Trust-Region Layers for Gaussian Policies

We present projections based on the three similarity measures, i.e., Frobenius norm, Wasserstein L2 distance, and KL divergence. These projections realize state-wise trust regions and can directly be integrated in the optimization process as differentiable neural network layers. Additionally, we extend the trust region layers to include an entropy constraint to gain control over the evolution of the policy entropy during optimization. The trust regions are defined by a distance or divergence $d(\pi(\cdot|s), \pi_{\text{old}}(\cdot|s))$ between probability distributions. Complementing Equation 1 with the trust region constraint leads to

$$\max_{\theta} \hat{J}(\pi_\theta, \pi_{\theta_{\text{old}}}) \quad \text{s.t.} \quad d(\pi_{\theta_{\text{old}}}(\cdot|s), \pi_\theta(\cdot|s)) \le \epsilon \quad \forall s \in \mathcal{S}. \tag{2}$$

While, in principle, we want to enforce the constraint for every possible state, in practice, we can only enforce them for states sampled from rollouts of the current policy.

To solve the problem in Equation 2, a standard neural network will output the parameters $\mu, \Sigma$ of a Gaussian distribution $\pi_\theta$, ignoring the trust region bounds. These parameters are provided to the trust region layers, together with the mean and covariance of the old policy and a parameter specifying the size of the trust region $\epsilon$. The new policy is then given by the output of the trust region layer. Since the old policy distribution is fixed, all distances or divergences used in this paper can be decomposed into a mean and a covariance dependent part. This enables us to use separate trust regions as well as bounds for mean and covariance, allowing for more flexibility in the algorithm. The trust region layers aim to project $\pi_\theta$ into the trust region by finding parameters $\tilde{\mu}$ and $\tilde{\Sigma}$ that are closest to the original parameters $\mu$ and $\Sigma$ while satisfying the trust region constraints. The projection is based on the same distance or divergence which was used to define the respective trust region. Formally, this corresponds to the following optimization problems for each $s$

$$\underset{\tilde{\mu}_s}{\arg\min}\, d_{\text{mean}}\left(\tilde{\mu}_s, \mu(s)\right), \quad \text{s.t.} \quad d_{\text{mean}}\left(\tilde{\mu}_s, \mu_{\text{old}}(s)\right) \le \epsilon_\mu, \quad \text{and} \tag{3}$$

$$\underset{\tilde{\Sigma}_s}{\arg\min}\, d_{\text{cov}}\left(\tilde{\Sigma}_s, \Sigma(s)\right), \quad \text{s.t.} \quad d_{\text{cov}}\left(\tilde{\Sigma}_s, \Sigma_{\text{old}}(s)\right) \le \epsilon_\Sigma, \tag{4}$$

where $\tilde{\mu}_s$ and $\tilde{\Sigma}_s$ are the optimization variables for state $s$. Here, $d_{\text{mean}}$ is the mean dependent part and $d_{\text{cov}}$ is the covariance dependent part of the employed distance or divergence. For brevity of notation we will neglect all dependencies on the state in the following. We denote the projected policy as $\tilde{\pi}(a|s) = \mathcal{N}(a|\tilde{\mu}, \tilde{\Sigma})$.

### 4.1 PROJECTION OF THE MEAN

For all three trust region objectives we make use of the same distance measure for the mean, the Mahalanobis distance. Hence, the optimization problem for the mean is given by

$$\arg\min_{\tilde{\mu}} (\mu - \tilde{\mu})^{\mathrm{T}} \Sigma_{\mathrm{old}}^{-1} (\mu - \tilde{\mu}) \quad \text{s.t.} \quad (\mu_{\mathrm{old}} - \tilde{\mu})^{\mathrm{T}} \Sigma_{\mathrm{old}}^{-1} (\mu_{\mathrm{old}} - \tilde{\mu}) \leq \epsilon_{\mu}. \tag{5}$$

By making use of the method of Lagrangian multipliers (see Appendix B.2), we can formulate the dual and solve it for the projected mean $\tilde{\mu}$ as

$$\tilde{\mu} = \frac{\mu + \omega\mu_{\mathrm{old}}}{1 + \omega} \quad \text{with} \quad \omega = \sqrt{\frac{(\mu_{\mathrm{old}} - \mu)^{\mathrm{T}} \Sigma_{\mathrm{old}}^{-1} (\mu_{\mathrm{old}} - \mu)}{\epsilon_{\mu}}} - 1. \tag{6}$$

This equation can directly be used as mean for the Gaussian policy, while it easily allows to compute gradients. Note, that for the mean part of the KL we would need to use the $\Sigma^{-1}$ instead of $\Sigma_{\mathrm{old}}^{-1}$ in the objective of Equation 5. Yet, this objective still results in a valid trust region problem which is much easier to optimize.

### 4.2 PROJECTION OF THE COVARIANCE

**Frobenius Projection.** The Frobenius projection formalizes the trust region for the covariance with the squared Frobenius norm of the matrix difference, which yields

$$\arg\min_{\tilde{\Sigma}} \mathrm{tr}\left((\Sigma - \tilde{\Sigma})^T (\Sigma - \tilde{\Sigma})\right), \quad \text{s.t.} \quad \mathrm{tr}\left((\Sigma_{\mathrm{old}} - \tilde{\Sigma})^T (\Sigma_{\mathrm{old}} - \tilde{\Sigma})\right) \leq \epsilon_{\Sigma}.$$

We again use the method of Lagrangian multipliers (see Appendix B.3) and get the covariance $\tilde{\Sigma}$ as

$$\tilde{\Sigma} = \frac{\Sigma + \eta\Sigma_{\mathrm{old}}}{1 + \eta} \quad \text{with} \quad \eta = \sqrt{\frac{\mathrm{tr}\left((\Sigma_{\mathrm{old}} - \Sigma)^T (\Sigma_{\mathrm{old}} - \Sigma)\right)}{\epsilon_{\Sigma}}} - 1, \tag{7}$$

where $\eta$ is the corresponding Lagrangian multiplier.

**Wasserstein Projection.** Deriving the Wasserstein projection follows the same procedure. We obtain the following optimization problem

$$\arg\min_{\tilde{\Sigma}} \mathrm{tr}\left(\Sigma_{\mathrm{old}}^{-1}\Sigma + \Sigma_{\mathrm{old}}^{-1}\tilde{\Sigma} - 2\Sigma_{\mathrm{old}}^{-1}\left(\Sigma^{1/2}\tilde{\Sigma}\Sigma^{1/2}\right)^{1/2}\right),$$

$$\text{s.t.} \quad \mathrm{tr}\left(\mathbb{I} + \Sigma_{\mathrm{old}}^{-1}\tilde{\Sigma} - 2\Sigma_{\mathrm{old}}^{-1}\left(\Sigma_{\mathrm{old}}^{1/2}\tilde{\Sigma}\Sigma_{\mathrm{old}}^{1/2}\right)^{1/2}\right) \leq \epsilon_{\Sigma}, \tag{8}$$

where $\mathbb{I}$ is the identity matrix. A closed form solution to this optimization problem can be found by using the methods outlined in Takatsu (2011). However, we found the resulting solution for the projected covariance matrices to be numerically unstable. Therefore, we made the simplifying assumption that both the current $\Sigma$ and the old covariance $\Sigma_{\mathrm{old}}$ commute with $\tilde{\Sigma}$. Under the common premise of diagonal covariances, this commutativity assumption always holds. For the more general case of arbitrary covariance matrices, we would need to ensure the matrices are sufficiently close together, which is effectively ensured by Equation 8. Again, we introduce Lagrange multipliers and solve the dual problem to obtain the optimal primal and dual variables (see Appendix B.4). Note however, that here we chose the *square root* of the covariance matrix[1] as primal variable. The corresponding projection for the square root covariance $\tilde{\Sigma}^{1/2}$ is then

$$\tilde{\Sigma}^{1/2} = \frac{\Sigma^{1/2} + \eta\Sigma_{\mathrm{old}}^{1/2}}{1 + \eta} \quad \text{with} \quad \eta = \sqrt{\frac{\mathrm{tr}\left(\mathbb{I} + \Sigma_{\mathrm{old}}^{-1}\Sigma - 2\Sigma_{\mathrm{old}}^{-1/2}\Sigma^{1/2}\right)}{\epsilon_{\Sigma}}} - 1, \tag{9}$$

where $\eta$ is the corresponding Lagrangian multiplier. We see the same pattern emerging as for the Frobenius projection. The chosen similarity measure reappears in the expression for the Lagrangian multiplier and the primal variables are weighted averages of the corresponding parameters of the old and the predicted Gaussian.

---

[1] We assume the true matrix square root $\Sigma = \Sigma^{1/2}\Sigma^{1/2}$ and not a Cholesky factor $\Sigma = LL^{\mathrm{T}}$ since it naturally appears in the expressions for the projected covariance from the original Wasserstein formulation.

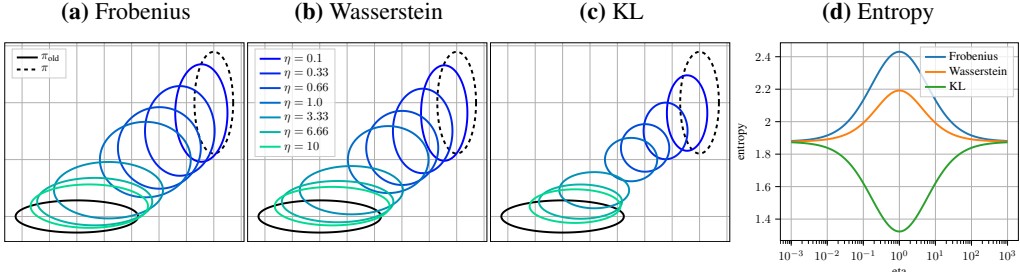

Figure 1: **(a)**, **(b)**, and **(c)**: Interpolated covariances for the different projections for various values of $\eta$. For Frobenius and Wasserstein the intermediate distributions clearly have a larger entropy, while for the KL projection the intermediate entropy is smaller. **(d)**: Entropy of the interpolated distributions. In this example $\pi$ and $\pi_{\text{old}}$ have the same entropy. It can be seen that the entropy increases for the Frobenius and Wasserstein projections when transitioning between the distributions, while it decreases for the KL. A more general statement regarding this can be found in Theorem 1.

**KL Projection.**    Identically to the previous two projections, we reformulate Equation 4 as

$$\underset{\tilde{\Sigma}}{\arg\min}\,\operatorname{tr}\left(\Sigma^{-1}\tilde{\Sigma}\right) + \log\frac{|\Sigma|}{|\tilde{\Sigma}|}, \quad \text{s.t.} \quad \operatorname{tr}\left(\Sigma_{\text{old}}^{-1}\tilde{\Sigma}\right) - d + \log\frac{|\Sigma_{\text{old}}|}{|\tilde{\Sigma}|} \le \epsilon_\Sigma, \tag{10}$$

where $d$ is the dimensionality of the action space. It is impossible to acquire a fully closed form solution for this problem. However, following Abdolmaleki et al. (2015), we can obtain the projected precision $\tilde{\Lambda} = \tilde{\Sigma}^{-1}$ by interpolation between the precision matrices of the old policy $\pi_{\text{old}}$ and the current policy $\pi$

$$\tilde{\Lambda} = \frac{\eta^* \Lambda_{\text{old}} + \Lambda}{\eta^* + 1}, \quad \eta^* = \underset{\eta}{\arg\min}\, g(\eta), \text{ s.t. } \eta \ge 0, \tag{11}$$

where $\eta$ is the corresponding Lagrangian multiplier and $g(\eta)$ the dual function. While this dual cannot be solved in closed form, an efficient solution exists using a standard numerical optimizer, such as BFGS, since it is a 1-D convex optimization. Regardless, we want a differentiable projection and thus also need to backpropagate the gradients through the numerical optimization. To this end, we follow Amos & Kolter (2017) and compute those gradients by taking the differentials of the KKT conditions of the dual. We refer to Appendix B.5 for more details and derivations.

**Entropy Control.**    Previous works (Akrour et al., 2019; Abdolmaleki et al., 2015) have shown the benefits of introducing an entropy constraint $\mathcal{H}(\pi_\theta) \ge \beta$ in addition to the trust region constraints. Such a constraint allows for more control over the exploration behavior of the policy. In order to endow our algorithm with this improved exploration behavior, we make use of the results from Akrour et al. (2019) and scale the standard deviation of the Gaussian distribution with a scalar factor $\exp\{(\beta - \mathcal{H}(\pi_\theta))/d\}$, which can also be individually computed per state.

### 4.3   ANALYSIS OF THE PROJECTIONS

It is instructive to compare the three projections. The covariance update is an interpolation for all three projections, but the quantities that are interpolated differ. For the Frobenius projection we directly interpolate between the old and current covariances (Equation 7), for the W2 projection between their respective matrix square roots (Equation 9), and for the KL projection between their inverses (Equation 11). In other words, each projection suggests which parametrization to use for the covariance matrix. The different interpolations also have an interesting effect on the entropy of the resulting covariances which can be observed in Figure 1. Further, we can prove the following theorem about the entropy of the projected distributions

**Theorem 1**  Let $\pi_\theta$ and $\pi_{\theta_{\text{old}}}$ be Gaussian and $\eta \ge 0$, then for the entropy of the projected distribution $\mathcal{H}(\tilde{\pi})$ it holds that $\mathcal{H}(\tilde{\pi}) \ge \text{minimum}(\mathcal{H}(\pi_\theta), \mathcal{H}(\pi_{\theta_{\text{old}}}))$ for the Frobenius (Equation 7) and the Wasserstein projection (Equation 9), as well as, $\mathcal{H}(\tilde{\pi}) \le \text{maximum}(\mathcal{H}(\pi_\theta), \mathcal{H}(\pi_{\theta_{\text{old}}}))$ for the KL projection (Equation 11).

Table 1: Mean return with 95% confidence interval of 20 epochs after completing 20% of the total training and for the last 20 epochs. We trained 40 different seeds for each experiment and computed five evaluation rollouts per epoch. The projections with (-E) and without entropy control are considered separately, therefore, each column may have up to two best runs (bold).

| | Hopper-v2 | | Walker2d-v2 | | Halfcheetah-v2 | | Ant-v2 | | Humanoid-v2 | |
| | 20% | final | 20% | final | 20% | final | 20% | final | 20% | final |
|---|---|---|---|---|---|---|---|---|---|---|
| FROB | $1646 \pm 19$ | $\mathbf{2578 \pm 17}$ | $2142 \pm 23$ | $3443 \pm 19$ | $2525 \pm 11$ | $3552 \pm 14$ | $1265 \pm 11$ | $3035 \pm 26$ | $2176 \pm 65$ | $5202 \pm 23$ |
| W2 | $1586 \pm 31$ | $2490 \pm 19$ | $2284 \pm 20$ | $3390 \pm 17$ | $\mathbf{2586 \pm 9}$ | $3692 \pm 15$ | $1362 \pm 12$ | $3086 \pm 28$ | $2502 \pm 87$ | $5057 \pm 25$ |
| KL | $1584 \pm 20$ | $2476 \pm 10$ | $2071 \pm 36$ | $\mathbf{3583 \pm 14}$ | $2369 \pm 9$ | $\mathbf{4255 \pm 17}$ | $1460 \pm 26$ | $\mathbf{3335 \pm 21}$ | $\mathbf{2923 \pm 53}$ | $\mathbf{5510 \pm 27}$ |
| PAPI | $1378 \pm 20$ | $\mathbf{2549 \pm 12}$ | $1663 \pm 21$ | $3232 \pm 20$ | $1875 \pm 5$ | $2380 \pm 6$ | $645 \pm 5$ | $3198 \pm 17$ | $1824 \pm 74$ | $5367 \pm 22$ |
| PPO-M | $1030 \pm 23$ | $2321 \pm 19$ | $1994 \pm 18$ | $2771 \pm 40$ | $1922 \pm 15$ | $3272 \pm 18$ | $1494 \pm 9$ | $2783 \pm 32$ | $604 \pm 10$ | $5172 \pm 23$ |
| PPO | $\mathbf{1881 \pm 30}$ | $2515 \pm 15$ | $\mathbf{2490 \pm 33}$ | $3447 \pm 17$ | $2048 \pm 8$ | $2880 \pm 8$ | $\mathbf{1657 \pm 10}$ | $2852 \pm 25$ | $1723 \pm 67$ | $4969 \pm 18$ |
| FROB-E | $1587 \pm 30$ | $2478 \pm 11$ | $2037 \pm 21$ | $3370 \pm 27$ | $\mathbf{2762 \pm 10}$ | $4568 \pm 14$ | $730 \pm 9$ | $\mathbf{3475 \pm 26}$ | $3662 \pm 44$ | $5807 \pm 18$ |
| W2-E | $1518 \pm 36$ | $2437 \pm 13$ | $2174 \pm 19$ | $3303 \pm 25$ | $2266 \pm 8$ | $4213 \pm 13$ | $855 \pm 27$ | $3361 \pm 30$ | $3658 \pm 56$ | $\mathbf{5844 \pm 8}$ |
| KL-E | $1502 \pm 18$ | $2497 \pm 16$ | $2215 \pm 31$ | $3171 \pm 28$ | $2611 \pm 12$ | $\mathbf{4584 \pm 18}$ | $955 \pm 16$ | $3437 \pm 21$ | $\mathbf{3801 \pm 42}$ | $5430 \pm 16$ |

The proof is based on the multiplicative version of the Brunn-Minkowski inequality and can be found in Appendix B.1. Intuitively, this implies that the Frobenius and Wasserstein projections act more *aggressively*, i. e., they rather yield a higher entropy, while the KL projection acts more *conservatively*, i. e., it rather yields a smaller entropy. This could also explain why many KL based trust region methods lose entropy too quickly and converge prematurely. By introducing an explicit entropy control, those effects can be mitigated.

## 4.4 Successive Policy Updates

The above projections can directly be implemented for training the current policy. Note, however, that at each epoch $i$ the policy $\pi_i$ predicted by the network before the projection layer does not respect the constraints and thus relies on calling this layer. The policy of the projection layer $\tilde{\pi}_i$ not only depends on the parameters of $\pi_i$ but also on the old policy network $\pi_{i,\text{old}} = \tilde{\pi}_{i-1}$. This would result in an ever-growing stack of policy networks becoming increasingly costly to evaluate. In other words, $\tilde{\pi}_i$ is computed using all stored networks of $\pi_i, \pi_{i-1}, \ldots, \pi_0$. We now discuss the parametrization of $\tilde{\pi}$ via amortized optimization.

We need to encode the information of the projection layer into the parameters $\theta$ of the next policy, i.e. $\tilde{\pi}(a|s; \theta) = p \circ \pi_\theta(a|s)$ is a composition function in which $p$ denotes the projection layer. The output of $\pi_\theta$ is $(\mu, \Sigma)$, and $p$ computes $(\tilde{\mu}, \tilde{\Sigma})$ according Equations 6, 7, 9, or 11. Formally, we aim to find a set of parameters $\theta^* = \arg\min_\theta \mathbb{E}_{\boldsymbol{s} \sim \rho_{\pi_{\text{old}}}} [d(\tilde{\pi}(\cdot|\boldsymbol{s}), \pi_\theta(\cdot|\boldsymbol{s}))]$, where $\rho_{\pi_{\text{old}}}$ is the state distribution of the old policy and $d$ is the similarity measure used for the projection, such that we minimize the expected distance or divergence between the projection and the current policy prediction.

The most intuitive way to solve this problem is to use the existing samples for additional regression steps after the policy optimization. Still, this adds a computational overhead. Therefore, we propose to concurrently optimize both objectives during training by penalizing the main objective, i. e.,

$$\arg\min_\theta \mathbb{E}_{(s,a) \sim \pi_{\theta_{\text{old}}}} \left[ \frac{\tilde{\pi}(a|s; \theta)}{\pi_{\theta_{\text{old}}}(a|s)} A^{\pi_{\text{old}}}(s, a) \right] - \alpha \mathbb{E}_{\boldsymbol{s} \sim p_{\pi_{\text{old}}}} [d(\tilde{\pi}(\cdot|\boldsymbol{s}; \theta), \pi_\theta(\cdot|\boldsymbol{s}))]. \quad (12)$$

Note that the importance sampling ratio is computed based on a Gaussian distribution generated by the trust region layer and not directly from the network output. Furthermore, the gradient for the regression penalty does not flow through the projection, it is solely acting as supervised learning signal. As appropriate similarity measures $d$ for the penalty, we resort to the measures used in each projection. For a detailed algorithmic view see Appendix A.

## 5 Experiments

**Mujoco Benchmarks** We evaluate the performance of our trust region layers regarding sample complexity and final reward in comparison to PAPI and PPO on the OpenAI gym benchmark suite (Brockman et al., 2016). We explicitly did not include TRPO in the evaluation, as Engstrom et al. (2020) showed that it can can achieve similar performance to PPO. For our experiments, the PAPI projection and its conservative PPO version are executed in the setting sent to us by the author. The hyperparameters for all three projections and PPO have been selected with Optuna (Akiba et al., 2019). See Appendix D for a full listing of all hyperparameters. We use a shared set of hyperparameters for all environments except for the Humanoid, which we optimized separately. Next

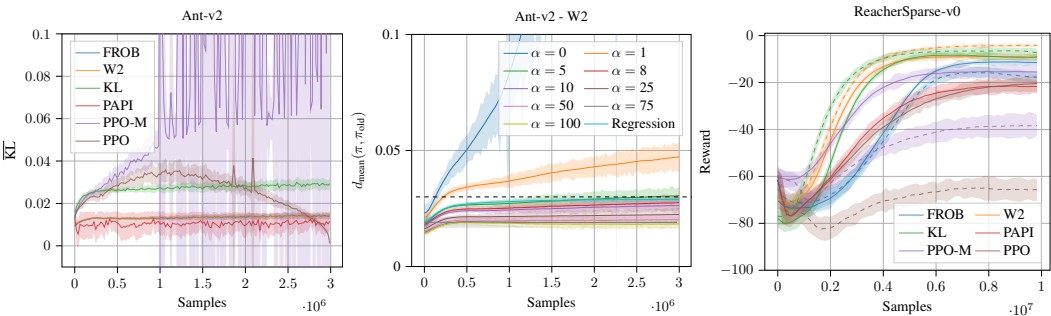

Figure 2: (Left): Mean KL divergence for Ant-v2 as a standardizing measure to compare the policy changes among all methods. (Center): Mahalanobis distance between the mean values of the unprojected and old policy when using different $\alpha$ in comparison to a full regression. The mean bound for the W2 projection is set to $0.03$ (dotted black line). (Right): Mean cumulative reward with $95\%$ confidence interval based on 40 seeds for the semi-sparse 5-link Reacher task. For each method, besides PAPI, we train policies with (dashed) and without (solid) contextual covariances.

to the standard PPO implementation with all code-level optimizations we further evaluate PPO-M, which only leverages the core PPO algorithm. Our projections and PPO-M solely use the observation normalization, network architecture, and initialization from the original PPO implementation. All algorithms parametrize the covariance as a non-contextual diagonal matrix. We refer to the Frobenius projection as *FROB*, the Wasserstein projection as *W2*, and the KL projection as *KL*.

Table 1 gives an overview of the final performance and convergence speed on the Mujoco benchmarks, Figure 4 in the appendix displays the full learning curves. After each epoch, we evaluate five episodes without applying exploration noise to obtain the return values. Note that we initially do not include the entropy projection to provide a fair comparison to PPO. The results show that our trust region layers are able to perform similarly or better than PPO and PAPI across all tasks. While the performance on the Hopper-v2 is comparable, the projections significantly outperform all baselines on the HalfCheetah-v2. The KL projection even demonstrates the best performance on the remaining three environments. Besides that, the experiments present a relatively balanced performance between projections, PPO, and PAPI. The differences are more apparent when comparing the projections to PPO-M, which uses the same implementation details as our projections. The asymptotic performance of PPO-M is on par for the Humanoid-v2, but it convergences much slower and is noticeably weaker on the remaining tasks. Consequently, the approximate trust region of PPO alone is not sufficient for good performance, only paired with certain implementation choices. Still, the original PPO cannot fully replace a mathematically sound trust region as ours, although it does not exhibit a strong performance difference. For this, Figure 2 visualizes the mean KL divergence at the end of each epoch for all methods. Despite the fact that neither W2 nor Frobenius projection use the KL, we leverage it here as a standardizing measure to compare the change in the policy distributions. All projections are characterized by an almost constant change, whereas for PPO-M the changes are highly inconsistent. The code-level optimizations of PPO can mitigate this to some extend but cannot properly enforce the desired constant change in the policy distribution. In particular, we have found that primarily the learning rate decay contributes to the relatively good behavior of PPO. Albeit, PAPI provides a similar principled trust region projection as we do, it still has some inconsistency by approaching the bound iteratively.

**Entropy Control**  To demonstrate the effect of combining our projections with entropy control, as described in Section 4.2, we evaluate all Mujoco tasks again for this extended setting. The target entropy in each iteration $i$ is computed by exponentially decaying the initial entropy $\mathcal{H}_0$ to $\kappa$ with temperature $\tau$ as $\kappa + (\mathcal{H}_0 - \kappa)\tau^{\frac{10i}{N}}$, where $N$ is the total number of training steps. The bottom of Table 1 shows the results for our projections with entropy control. Especially on the more complex tasks with more exploration, all three projections significantly benefit from the entropy control. Their asymptotic performance for the HalfCheetah-v2, Ant-v2, and Humanoid-v2 increases and yields a much faster convergence in the latter. For the other Mujoco tasks the performance remains largely constant since the complexity of these tasks is insufficient to benefit from an explicit entropy control, as also noted by Pajarinen et al. (2019) and Abdolmaleki et al. (2015).

**Contextual Covariances.** To emphasize the advantage of *state-wise* trust regions we consider the case of policies with state-dependent covariances. Existing methods, such as PPO and TRPO, are rarely used in this setting. In addition, PAPI cannot project the covariance in the contextual case. Further, Andrychowicz et al. (2020) demonstrated that for the standard Mujoco benchmarks, contextual covariances are not beneficial in an on-policy setting. Therefore, we choose to evaluate on a task motivated from optimal control which benefits from a contextual covariance. We extend the Mujoco *Reacher-v2* to a 5-link planar robot, the distance penalty to the target is only provided in the last time step, $t = 200$, and the observation space also contains the current time step $t$. This semi-sparse reward specification imposes a significantly harder exploration problem as the agent is only provided with a feedback at the last time step. We again tuned all hyperparameters using Optuna Akiba et al. (2019) and did not include the entropy projection. All feasible approaches are compared with and without contextual covariances, the results therefor are presented in Figure 2 (right). All three projections significantly outperform the baseline methods with the non-contextual covariance. Additionally, both the W2 and KL projection improve their results in the contextual case. In contrast, all baselines decrease in performance and are not able to leverage the advantage of contextual information. This poor performance mainly originates from incorrect exploitation. PPO reduces the covariance too quickly, whereas PAPI reduces it too slowly, leading to a suboptimal performance for both. The Frobenius projection, however, does not benefit from contextual covariances either, since numerical instabilities arise from too small covariance values close to convergence. Those issues can be mitigated using a smaller covariance bound, but they cannot be entirely avoided. The KL projection, while yielding the best results throughout all experiments, relies on a numerical optimization. Generally, this is computationally expensive, however, by leveraging an efficient C++ implementation this problem can be negated (see Appendix B.5). As a bonus, the KL projection has all properties of existing KL-based trust region methods that have monotonic improvement guarantees. Nevertheless, for quick benchmarks, the W2 is preferred, given it is slightly less prone to hyperparameter choices and does not require a dedicated custom implementation.

**Trust Region Regression Loss.** Lastly, we investigate the main approximation of our approach, the trust region regression loss (Equation 12). In the following ablation, we evaluate how different choices of the regression weight $\alpha$ affect the constraint satisfaction. Figure 2 (center) shows the Mahalanobis distance between the unprojected and the old policy means for different $\alpha$ values. In addition, for one run we choose $\alpha = 0$ and execute the trust region regression separately after each epoch for several iterations. One key observation is that decreasing the penalty up to a certain threshold leads to larger changes in the policy and pushes the mean closer to its maximum bound. Intuitively, this can be explained by the construction of the bound. As the penalty is added only to the loss when the bound is violated, larger changes in the policy are punished while smaller steps do not directly affect the loss negatively. By selecting a larger $\alpha$, this behavior is reinforced. Furthermore, we can see that some smaller values of $\alpha$ yield a behavior which is similar to the full regression setting. Consequently, it is justified to use a computationally simpler penalty instead of performing a full regression after each epoch.

## 6 Discussion and Future Work

In this work we proposed differentiable projection layers to enforce trust region constraints for Gaussian policies in deep reinforcement learning. While being more stable than existing methods, they also offer the benefit of imposing the constraints on a state level. Unlike previous approaches that only constrain the expected change between successive policies and for whom monotonic improvement guarantees thus only hold approximately, we can constrain the maximum change. Our results illustrate that trust regions are an effective tool in policy search for a wide range of different similarity measures. Apart from the commonly used reverse KL, we also leverage the Wasserstein distance and Frobenius norm. We demonstrated the subtle but important differences between those three different types of trust regions and showed our benchmark performance is on par or better than existing methods that use more code-level optimizations. For future work, we plan to continue our research with more exploration-heavy environments, in particular with contextual covariances. Additionally, more sophisticated heuristics or learning methods could be used to adapt the trust region bounds for better performance. Lastly, we are interested in using our trust region layers for other deep reinforcement learning approaches, such as actor-critic methods.

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

## A  ALGORITHM

---

**Algorithm 1** Differentiable Trust Region Layer. The trust region layer acts as final layer after predicting a Gaussian distribution. It projects this predicted Gaussian onto the trust region in case it is violating the specified bounds. As output it generates a projected mean and covariance that satisfy the respective trust region bound. The entropy control in the last step can be disabled.

Initialize bounds $\epsilon_\mu, \epsilon_\Sigma$, temperature $\tau$ as well as target $\kappa$ and initial entropy $\mathcal{H}_0$.

1: **procedure** TRUSTREGIONLAYER($\mu, \Sigma, \mu_{\text{old}}, \Sigma_{\text{old}}$)
2:     **if** $d_{\text{mean}}(\mu, \mu_{\text{old}}) > \epsilon_\mu$ **then**
3:         Compute $\tilde{\mu}$ with Equation 6
4:     **else**
5:         $\tilde{\mu} = \mu$
6:     **if** $d_{\text{cov}}(\Sigma, \Sigma_{\text{old}}) > \epsilon_\Sigma$ **then**
7:         Compute $\tilde{\Sigma}$ with Equations 7, 9, or 11
8:     **else**
9:         $\tilde{\Sigma} = \Sigma$
10:     $\beta = \kappa + (\mathcal{H}_0 - \kappa)\tau^{\frac{10i}{N}}$          ▷ (Optional) entropy control as described in Section 4.2
11:     **if** $\mathcal{H}(\Sigma) < \beta$ **then**
12:         $c = \exp\left\{(\beta - \mathcal{H}(\Sigma))/\dim(a)\right\}$
13:         $\tilde{\Sigma} = c\tilde{\Sigma}$
14:     **return** $\tilde{\mu}, \tilde{\Sigma}$

---

**Algorithm 2** Algorithmic view of the proposed Trust Region Projections. The trust region projections itself do not require approximations, the old policy update in the last step is the only point where we introduce an approximation. This update would normally require additional supervised regression steps that minimize the distance between the network output and the projection. However, by leveraging the regression penalty during policy optimization this optimization step can be omitted. Both approaches yield a policy, which is independent of the old policy distribution, i.e. it can act without the projection while maintaining the trust region. However, the penalty does not require additional computation and the policy can directly generate new trajectories, equivalently to other trust region methods, such as PPO.

---

1: Initialize policy $\theta_{0,0}$
2: **for** $i = 0, 1, \ldots, N$ **do**          ▷ epoch
3:     Collect set of trajectories $\mathcal{D}_i = \{\tau_k\}$ with $\pi(\theta_{i,0})$
4:     Compute advantage estimates $\hat{A}_t$ with GAE
5:     **for** $j = 0, 1, \ldots, M$ **do**
6:         Use $\pi(\theta_{i,j})$ to predict Gaussian action distributions $\mathcal{N}(\mu_{i,j}, \Sigma_{i,j})$ for $\mathcal{D}_i$
7:         $\tilde{\pi} = $ TRUSTREGIONLAYER($\mu_{i,j}, \Sigma_{i,j}, \mu_{i,0}, \Sigma_{i,0}$)
8:         Update policy with Adam using the following policy gradient:

$$\theta_{i,j+1} \leftarrow \text{Adam}\left(\nabla_\theta \left[\mathbb{E}_{\pi(\theta_{i,0})}\left[\frac{\tilde{\pi}(a|s;\theta)}{\pi(a|s;\theta_{i,0})}\hat{A}_t\right] - \alpha\mathbb{E}_{s\sim p_{\pi(\theta_{i,0})}}\left[d\left(\tilde{\pi}(\cdot|s;\theta), \pi(\cdot|s;\theta)\right)\right]\right]\Big|_{\theta=\theta_{i,j}}\right)$$

9:     Successive policy update: $\theta_{i+1,0} \leftarrow \theta_{i,M}$

---

## B  DERIVATIONS

### B.1  PROOF OF THEOREM 1

This section provides a proof for Theorem 1. We mainly used the multiplicative version of the Brunn-Minkowski inequality

$$\log(\alpha|\Sigma_1| + \beta|\Sigma_2|) \geq \log(|\Sigma_1|)^\alpha \log(|\Sigma_2|)^\beta$$

where $\Sigma_1, \Sigma_2$ are p.s.d, $\alpha, \beta$ are positive, and $\alpha + \beta = 1$.

**Frobenius Projection**

$$
\begin{aligned}
\mathrm{H}(\tilde{\pi}) &= 0.5 \log |2\pi e \tilde{\Sigma}| \\
&= 0.5 \log \left| 2\pi e \left( \frac{1}{\eta+1}\Sigma + \frac{\eta}{\eta+1}\Sigma_{\text{old}} \right) \right| \\
&\geq 0.5 \log \left| (2\pi e \Sigma)^{\frac{1}{\eta+1}} \det (2\pi e \Sigma_{\text{old}})^{\frac{\eta}{\eta+1}} \right| \\
&= \frac{1}{\eta+1} 0.5 \log |2\pi e \Sigma| + \frac{\eta}{\eta+1} 0.5 \log |2\pi e \Sigma_{\text{old}}| \\
&= \frac{1}{\eta+1} \mathrm{H}(\pi_\theta) + \frac{\eta}{\eta+1} \mathrm{H}(\pi_{\theta_{\text{old}}}) \geq \text{minimum} \left( \mathrm{H}(\pi_\theta), \mathrm{H}(\pi_{\theta_{\text{old}}}) \right)
\end{aligned}
$$

**Wasserstein Projection**   Let $k$ denote the dimensionality of the distributions under consideration.

$$
\begin{aligned}
\mathrm{H}(\tilde{\pi}) &= 0.5 \log (2\pi e)^k |\tilde{\Sigma}| \\
&= 0.5 \log (2\pi e)^k \left| \left( \frac{1}{\eta+1}\Sigma^{0.5} + \frac{\eta}{\eta+1}\Sigma_{\text{old}}^{0.5} \right) \right|^2 \\
&= 0.5 \log (2\pi e)^k + \log \left| \frac{1}{\eta+1}\Sigma^{0.5} + \frac{\eta}{\eta+1}\Sigma_{\text{old}}^{0.5} \right| + \log \left| \frac{1}{\eta+1}\Sigma^{0.5} + \frac{\eta}{\eta+1}\Sigma_{\text{old}}^{0.5} \right| \\
&\geq 0.5 \log (2\pi e)^k + \log \left| \Sigma^{0.5} \right|^{\frac{1}{\eta+1}} \left| \Sigma_{\text{old}}^{0.5} \right|^{\frac{\eta}{\eta+1}} + \log \left| \Sigma^{0.5} \right|^{\frac{1}{\eta+1}} \left| \Sigma_{\text{old}}^{0.5} \right|^{\frac{\eta}{\eta+1}} \\
&= 0.5 \log (2\pi e)^k + \log |\Sigma|^{\frac{1}{\eta+1}} |\Sigma_{\text{old}}|^{\frac{\eta}{\eta+1}} \\
&= 0.5 \log \left( |(2\pi e \Sigma|^{\frac{1}{\eta+1}} |2\pi e \Sigma_{\text{old}}|^{\frac{\eta}{\eta+1}} \right) \\
&= \frac{1}{\eta+1} 0.5 \log |2\pi e \Sigma| + \frac{\eta}{\eta+1} 0.5 \log |2\pi e \Sigma_{\text{old}}| \\
&= \frac{1}{\eta+1} \mathrm{H}(\pi_\theta) + \frac{\eta}{\eta+1} \mathrm{H}(\pi_{\theta_{\text{old}}}) \geq \text{minimum} \left( \mathrm{H}(\pi), \mathrm{H}(\pi_{\text{old}}) \right)
\end{aligned}
$$

**KL Projection**

$$
\begin{aligned}
\mathrm{H}(\tilde{\pi}) &= 0.5 \log |2\pi e \tilde{\Sigma}| \\
&= 0.5 \log \left| \frac{1}{\eta+1}(2\pi e \Sigma)^{-1} + \frac{\eta}{\eta+1}(2\pi e \Sigma_{\text{old}})^{-1} \right|^{-1} \\
&= -0.5 \log \left| \frac{1}{\eta+1}(2\pi e \Sigma)^{-1} + \frac{\eta}{\eta+1}(2\pi e \Sigma_{\text{old}})^{-1} \right| \\
&\leq -0.5 \log \left( \left| (2\pi e \Sigma)^{-1} \right|^{\frac{1}{\eta+1}} \left| (2\pi e \Sigma_{\text{old}})^{-1} \right|^{\frac{\eta}{\eta+1}} \right) \\
&= 0.5 \log \left( |2\pi e \Sigma|^{\frac{1}{\eta+1}} |2\pi e \Sigma_{\text{old}}|^{\frac{\eta}{\eta+1}} \right) \quad \text{(use the fact that: } \det(A^{-1}) = 1/\det(A)) \\
&= \frac{1}{\eta+1} 0.5 \log |2\pi e \Sigma| + \frac{\eta}{\eta+1} 0.5 \log |2\pi e \Sigma_{\text{old}}| \\
&= \frac{1}{\eta+1} \mathrm{H}(\pi_\theta) + \frac{\eta}{\eta+1} \mathrm{H}(\pi_{\theta_{\text{old}}}) \leq \text{maximum} \left( \mathrm{H}(\pi_\theta), \mathrm{H}(\pi_{\text{old}}) \right)
\end{aligned}
$$

## B.2   MEAN PROJECTION

First, we consider only the mean objective

$$
\min_{\tilde{\mu}} \quad (\mu - \tilde{\mu})^{\mathrm{T}} \Sigma_{\text{old}}^{-1} (\mu - \tilde{\mu})
$$

$$
\text{s.t.} \quad (\mu_{\text{old}} - \tilde{\mu})^{\mathrm{T}} \Sigma_{\text{old}}^{-1} (\mu_{\text{old}} - \tilde{\mu}) \leq \epsilon_\mu,
$$

which give us the following dual

$$
\mathcal{L}(\tilde{\mu}, \omega) = (\mu - \tilde{\mu})^{\mathrm{T}} \Sigma_{\text{old}}^{-1} (\mu - \tilde{\mu}) + \omega \left( (\mu_{\text{old}} - \tilde{\mu})^{\mathrm{T}} \Sigma_{\text{old}}^{-1} (\mu_{\text{old}} - \tilde{\mu}) - \epsilon_\mu \right). \tag{13}
$$

Differentiating w.r.t. $\tilde{\mu}$ yields

$$\frac{\partial \mathcal{L}(\tilde{\mu}, \omega)}{\partial \tilde{\mu}} = 2\Sigma_{\text{old}}^{-1} (\tilde{\mu} - \mu) - 2\omega \Sigma_{\text{old}}^{-1} (\tilde{\mu} - \mu_{\text{old}}).$$

Setting the derivative to $0$ and solving for $\tilde{\mu}$ gives

$$\tilde{\mu}^* = \frac{\mu + \omega \mu_{old}}{1 + \omega}.$$

Inserting the optimal mean $\tilde{\mu}^*$ in Equation 13 results in

$$L(\omega) = \left( \frac{\mu + \omega \mu_{old}}{1 + \omega} - \mu \right)^T \Sigma_{\text{old}}^{-1} \left( \frac{\mu + \omega \mu_{old}}{1 + \omega} - \mu \right) +$$

$$+ \omega \left( \left( \frac{\mu + \omega \mu_{old}}{1 + \omega} - \mu_{old} \right)^T \Sigma_{\text{old}}^{-1} \left( \frac{\mu + \omega \mu_{old}}{1 + \omega} - \mu_{old} \right) - \epsilon_\mu \right)$$

$$= \frac{\omega^2 (\mu - \mu_{old})^T \Sigma_{\text{old}}^{-1} (\mu - \mu_{old})}{(1 + \omega)^2} + \frac{\omega (\mu - \mu_{old})^T \Sigma_{\text{old}}^{-1} (\mu - \mu_{old})}{(1 + \omega)^2} - \omega \epsilon_\mu.$$

Thus, differentiating w.r.t $\omega$ yields

$$\frac{\partial \mathcal{L}(\omega)}{\partial \omega} = \frac{(\mu - \mu_{old})^T \Sigma_{\text{old}}^{-1} (\mu - \mu_{old})}{(1 + \omega)^2} - \epsilon_\mu.$$

Now solving $\frac{\partial \mathcal{L}(\omega)}{\partial \omega} \overset{!}{=} 0$ for $\omega$, we arrive at

$$\omega^* = \sqrt{\frac{(\mu - \mu_{old})^T \Sigma_{\text{old}}^{-1} (\mu - \mu_{old})}{\epsilon_\mu}} - 1.$$

## B.3 FROBENIUS COVARIANCE PROJECTION

We consider the following objective for the covariance part

$$\min_{\tilde{\Sigma}} \quad \left\| \tilde{\Sigma} - \Sigma \right\|_F^2$$

$$\text{s.t.} \quad \left\| \tilde{\Sigma} - \Sigma_{\text{old}} \right\|_F^2 \leq \epsilon_\Sigma$$

with the corresponding Lagrangian

$$\mathcal{L}(\tilde{\Sigma}, \eta) = \left\| \tilde{\Sigma} - \Sigma \right\|_F^2 + \eta \left( \left\| \tilde{\Sigma} - \Sigma_{\text{old}} \right\|_F^2 - \epsilon_\Sigma \right). \tag{14}$$

Differentiating w.r.t. $\tilde{\Sigma}$ yields

$$\frac{\partial \mathcal{L}(\tilde{\Sigma}, \eta)}{\partial \tilde{\Sigma}} = 2 \left( \left( \tilde{\Sigma} - \Sigma \right) + \eta \left( \Sigma_{\text{old}} - \Sigma \right) \right).$$

We can again solve for $\tilde{\Sigma}$ by setting the derivative to $0$, i.e.,

$$\tilde{\Sigma}^* = \frac{\Sigma + \eta \Sigma_{\text{old}}}{1 + \eta}.$$

Inserting $\tilde{\Sigma}^*$ into Equation 14 yields the dual function

$$g(\eta) = \left\| \frac{\Sigma + \eta \Sigma_{\text{old}}}{1 + \eta} - \Sigma \right\|_F^2 + \eta \left( \left\| \frac{\Sigma + \eta \Sigma_{\text{old}}}{1 + \eta} - \Sigma_{\text{old}} \right\|_F^2 - \epsilon_\Sigma \right).$$

Differentiating w.r.t. $\eta$ results in

$$\frac{\partial \mathcal{L}(\eta)}{\partial \eta} = \frac{\left\| \Sigma - \Sigma_{\text{old}} \right\|_F^2}{(1 + \eta)^2} - \epsilon_\Sigma.$$

Hence, $\frac{\partial \mathcal{L}(\eta)}{\partial \eta} \overset{!}{=} 0$ yields

$$\eta^* = \frac{\left\| \Sigma - \Sigma_{\text{old}} \right\|_F}{\sqrt{\epsilon_\Sigma}} - 1.$$

## B.4 Wasserstein Covariance Projection

As described in the main text, the Gaussian distributions to have been rescaled by $\Sigma_{\mathrm{old}}^{-1}$ to measure the distance in the metric space that is defined by the variance of the data. For notational simplicity, we show the derivation of the covariance projection only for the unscaled scenario. The scaled version can be obtained by a simple redefinition of the covariance matrices. For our covariance projection we are interested in solving the following optimization problem

$$\min_{\tilde{\Sigma}} \quad \mathrm{tr}\left(\tilde{\Sigma} + \Sigma - 2\left(\Sigma^{1/2}\tilde{\Sigma}\Sigma^{1/2}\right)^{1/2}\right)$$

$$\text{s.t.} \quad \mathrm{tr}\left(\tilde{\Sigma} + \Sigma_{\mathrm{old}} - 2\left(\Sigma_{\mathrm{old}}^{1/2}\tilde{\Sigma}\Sigma_{\mathrm{old}}^{1/2}\right)^{1/2}\right) \leq \epsilon_{\Sigma},$$

which leads to the following Lagrangian function

$$\mathcal{L}(\tilde{\Sigma}, \eta) = \mathrm{tr}\left(\tilde{\Sigma} + \Sigma - 2\left(\Sigma^{1/2}\tilde{\Sigma}\Sigma^{1/2}\right)^{1/2}\right)$$
$$+ \eta\left(\mathrm{tr}\left(\tilde{\Sigma} + \Sigma_{\mathrm{old}} - 2\left(\Sigma_{\mathrm{old}}^{1/2}\tilde{\Sigma}\Sigma_{\mathrm{old}}^{1/2}\right)^{1/2}\right) - \epsilon_{\Sigma}\right). \tag{15}$$

Assuming that $\tilde{\Sigma}$ commutes with $\Sigma$ as well as $\Sigma_{\mathrm{old}}$, Equation 15 simplifies to

$$\mathcal{L}(\tilde{\Sigma}, \eta) = \mathrm{tr}\left(\tilde{\Sigma} + \Sigma - 2\tilde{\Sigma}^{1/2}\Sigma^{1/2}\right) + \eta\left(\mathrm{tr}\left(\tilde{\Sigma} + \Sigma_{\mathrm{old}} - 2\tilde{\Sigma}^{1/2}\Sigma_{\mathrm{old}}^{1/2}\right) - \epsilon\right)$$
$$= \mathrm{tr}\left(S^2 + \Sigma - 2S\Sigma^{1/2}\right) + \eta\left(\mathrm{tr}\left(S^2 + \Sigma_{\mathrm{old}} - 2S\Sigma_{\mathrm{old}}^{1/2}\right) - \epsilon\right), \tag{16}$$

where $S$ is the unique positive semi-definite root of the positive semi-definite matrix $\tilde{\Sigma}$, i.e. $S = \tilde{\Sigma}^{1/2}$. Instead of optimizing the objective w.r.t $\tilde{\Sigma}$, we optimize w.r.t $S$ in order, which greatly simplifies the calculation. That is, we solve

$$\frac{\partial \mathcal{L}(S, \eta)}{\partial S} = (1+\eta)2S - 2\left(\Sigma^{1/2} + \eta\Sigma_{\mathrm{old}}^{1/2}\right) \stackrel{!}{=} 0$$

for $S$, which leads us to

$$S^* = \frac{\Sigma^{1/2} + \eta\Sigma_{\mathrm{old}}^{1/2}}{1+\eta}, \tilde{\Sigma}^* \qquad = \frac{\Sigma + \eta^2\Sigma_{\mathrm{old}} + 2\eta\Sigma^{1/2}\Sigma_{\mathrm{old}}^{1/2}}{(1+\eta)^2}.$$

Inserting this into Equation 16 yields the dual function

$$g(\eta) = \frac{\eta\left(\mathrm{tr}\left(\Sigma + \Sigma_{\mathrm{old}} - 2\Sigma^{1/2}\Sigma_{\mathrm{old}}^{1/2}\right)\right)}{1+\eta} - \eta\epsilon_{\Sigma}$$

The derivative of the dual w.r.t. $\eta$ is given by

$$\frac{\partial \mathcal{L}(\eta)}{\partial \eta} = \frac{\mathrm{tr}\left(\tilde{\Sigma} + \Sigma_{\mathrm{old}} - 2\tilde{\Sigma}^{1/2}\Sigma_{\mathrm{old}}^{1/2}\right)}{(1+\lambda)^2} - \epsilon_{\Sigma}.$$

Now solving $\frac{\partial \mathcal{L}(\eta)}{\partial \eta} \stackrel{!}{=} 0$ for $\eta$, we arrive at

$$\eta^* = \sqrt{\frac{\mathrm{tr}\left(\tilde{\Sigma} + \Sigma_{\mathrm{old}} - 2\tilde{\Sigma}^{1/2}\Sigma_{\mathrm{old}}^{1/2}\right)}{\epsilon_{\Sigma}}} - 1$$

## B.5 KL-Divergence Projection

We derive the KL-Divergence projection in its general form, i.e., simultaneous projection of mean and covariance under an additional entropy constraint

$$\tilde{\pi}^* = \arg\min_{\tilde{\pi}} \mathrm{KL}\left(\tilde{\pi}||\pi_{\theta}\right) \quad \text{s.t.} \quad \mathrm{KL}\left(\tilde{\pi}||\pi_{\theta_{\mathrm{old}}}\right) \leq \epsilon, \quad \mathrm{H}\left(\tilde{\pi}\right) \geq \beta.$$

Instead of working with this minimization problem we consider the equivalent maximization problem

$$\tilde{\pi}^* = \arg\max_{\tilde{\pi}} -\mathrm{KL}\left(\tilde{\pi}||\pi_\theta\right) \quad \text{s.t.} \quad \mathrm{KL}\left(\tilde{\pi}||\pi_{\theta_{\mathrm{old}}}\right) \leq \epsilon, \quad \mathrm{H}\left(\tilde{\pi}\right) \geq \beta, \tag{17}$$

which is similar to the one considered in *Model Based Relative Entropy Stochastic Search* (MORE) (Abdolmaleki et al., 2015), with a few distinctions. To see those distinctions let $\eta$ and $\omega$ denote the Lagrangian multipliers corresponding to the KL and entropy constraint respectively and consider the Lagrangian corresponding to the optimization problem in Equation 17

$$\mathcal{L} = -\mathrm{KL}(\tilde{\pi}||\pi_\theta) + \eta\left(\epsilon - \mathrm{KL}(\tilde{\pi}||\pi_{\theta_{\mathrm{old}}})\right) + \omega\left(\mathrm{H}(\tilde{\pi}) - \beta\right)$$
$$= \mathbb{E}_{\tilde{\pi}}\left[\log\pi_\theta\right] + \eta\left(\epsilon - \mathrm{KL}(\tilde{\pi}||\pi_{\theta_{\mathrm{old}}})\right) + (\omega + 1)\mathrm{H}(\tilde{\pi}) - \omega\beta.$$

Opposed to Abdolmaleki et al. (2015) we are not working with an unknown reward but using the log density of the target distribution $\pi$ instead. Thus we do not need to fit a surrogate and can directly read off the parameters of the squared reward. They are given by the natural parameters of $\pi$, i.e, $\Lambda = \Sigma^{-1}$ and $q = \Sigma^{-1}\mu$. Additionally, we need to add a constant 1 to $\omega$ to account for the additional entropy term in the original objective, similar to (Arenz et al., 2018).

Following the derivations from Abdolmaleki et al. (2015) and Arenz et al. (2018) we can obtain a closed form solution for the natural parameters of $\tilde{\pi}$, given the Lagrangian multipliers $\eta$ and $\omega$

$$\tilde{\Lambda} = \frac{\eta\Lambda_{\mathrm{old}} + \Lambda}{\eta + 1 + \omega} \quad \text{and} \quad \tilde{q} = \frac{\eta q_{\mathrm{old}} + q}{\eta + 1 + \omega}. \tag{18}$$

To obtain the optimal Lagrangian multipliers we can solve the following convex dual function using gradient descent

$$g(\eta, \omega) = \eta\epsilon - \omega\beta + \eta\left(-\frac{1}{2}q_{\mathrm{old}}^T\Lambda_{\mathrm{old}}^{-1}q_{\mathrm{old}} + \frac{1}{2}\log\det\left(\Lambda\right) - \frac{k}{2}\log(2\pi)\right)$$
$$+ (\eta + 1 + \omega)\left(\frac{1}{2}\tilde{q}^T\tilde{\Lambda}^{-1}q - \frac{1}{2}\log\det\left(\tilde{\Lambda}\right) + \frac{k}{2}\log(2\pi)\right) + \mathrm{const},$$
$$\frac{\partial g(\eta, \omega)}{\partial\eta} = \epsilon - \mathrm{KL}(\tilde{\pi}||\pi_{\theta_{\mathrm{old}}}) \quad \text{and} \quad \frac{\partial g(\eta, \omega)}{\partial\omega} = \mathrm{H}(\tilde{\pi}) - \beta.$$

Given the optimal Lagrangian multipliers, $\eta^*$ and $\omega^*$ we obtain the parameters of the optimal distribution $\tilde{\pi}^*$ using Equation 18.

**Forward Pass**   For the forward pass we compute the natural parameters of $\pi$, solve the optimization problem and compute mean and covariance of $\tilde{\pi}^*$ from the optimal natural parameters. The corresponding compute graph is given in Figure 3.

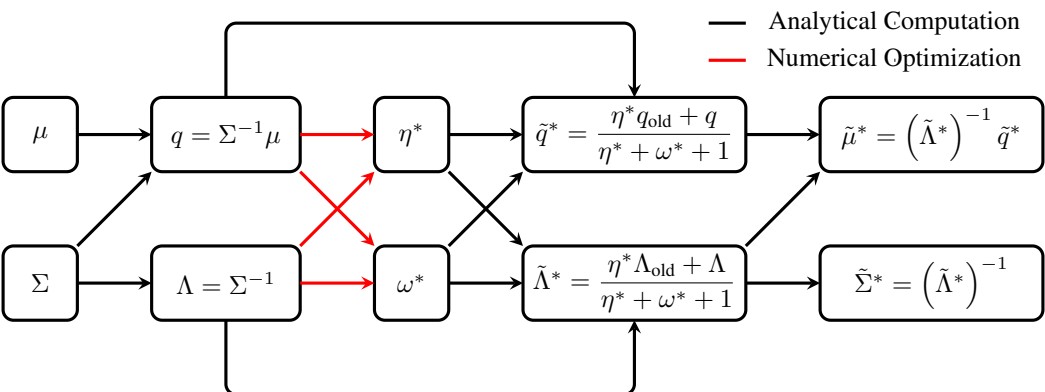

Figure 3: Compute graph of the KL projection layer. The layer first computes the natural parameters of $\pi$ from the mean and covariance. Then it numerically optimizes the dual to obtain the optimal Lagrangian multipliers which are used to get the optimal natural parameters. Ultimately, the optimal mean and covariance are computed from the optimal natural parameters. We omit the dependency on constants, i.e., the bound $\epsilon$ and $\beta$ as well as the parameters of $\pi_{\mathrm{old}}$ for clarity of the visualization.

**Backward Pass** Given the computational graph in Figure 3 gradients can be propagated back though the layer using standard back-propagation. All gradients for the analytical computations (black arrows in Figure 3) are straight forward and can be found in (Petersen & Pedersen, 2012). For the gradients of the numerical optimization of the dual (red arrows in Figure 3) we follow Amos & Kolter (2017) and differentiate the KKT conditions around the optimal Lagrangian multipliers computed during the forward pass. The KKT Conditions of the dual are given by

$$\nabla g(\eta^*, \omega^*) + m^T \nabla \begin{pmatrix} -\eta^* \\ -\omega^* \end{pmatrix} = \begin{pmatrix} \epsilon - \mathrm{KL}\left(\tilde{\pi}^* || \pi_{\theta_{\mathrm{old}}}\right) - m_1 \\ \mathrm{H}(\tilde{\pi}^*) - \beta - m_2 \end{pmatrix} = 0, \quad \text{(Stationarity)}$$

$$m_1(-\eta^*) = 0 \quad \text{and} \quad m_2(-\omega^*) = 0 \quad \text{(Complementary Slackness)}$$

here $m = (m_1, m_2)^T$ denotes the Lagrangian multipliers for the box constraints of the dual ($\eta$ and $\omega$ need to be non-negative). Taking the differentials of those conditions yields the equation system

$$\begin{pmatrix} -\dfrac{\partial \mathrm{KL}\left(\tilde{\pi}^* || \pi_{\theta_{\mathrm{old}}}\right)}{\partial \eta^*} & -\dfrac{\partial \mathrm{KL}\left(\tilde{\pi}^* || \pi_{\theta_{\mathrm{old}}}\right)}{\partial \omega^*} & -1 & 0 \\ \dfrac{\partial \mathrm{H}(\tilde{\pi}^*)}{\partial \eta^*} & \dfrac{\partial \mathrm{H}(\tilde{\pi}^*)}{\partial \omega^*} & 0 & -1 \\ -m_1 & 0 & -\eta^* & 0 \\ 0 & -m_2 & 0 & -\omega^* \end{pmatrix} \begin{pmatrix} d\eta \\ d\omega \\ dm_1 \\ dm_2 \end{pmatrix}$$

$$= \begin{pmatrix} \dfrac{\partial \mathrm{KL}\left(\tilde{\pi}^* || \pi_{\theta_{\mathrm{old}}}\right)}{\partial q} dq + \dfrac{\partial \mathrm{KL}\left(\tilde{\pi}^* || \pi_{\theta_{\mathrm{old}}}\right)}{\partial \Lambda} d\Lambda \\ -\dfrac{\partial \mathrm{H}(\tilde{\pi}^*)}{\partial q} dq - \dfrac{\partial \mathrm{H}(\tilde{\pi}^*)}{\partial \Lambda} d\Lambda \\ 0 \\ 0 \end{pmatrix}$$

which is (analytically) solved to obtain the desired partial derivatives

$$\frac{\partial \eta}{\partial q}, \frac{\partial \eta}{\partial \Lambda}, \frac{\partial \omega}{\partial q} \quad, \text{ and } \quad \frac{\partial \omega}{\partial \Lambda}.$$

**Implementation** We implemented the whole layer using C++, Armadillo, and OpenMP for parallelization. The implementation saves all necessary quantities for the backward pass and thus a numerical optimization is only necessary during the forward pass. Before we perform a numerical optimization we check whether it is actually necessary. If the target distribution $\tilde{\pi}$ is within the trust region we immediately can set $\tilde{\pi}^* = \pi_\theta$, i.e., the forward and backward pass become the identity mapping. This check yield significant speed-ups, especially in early iterations, if the target is still close to the old distribution. If the projection is necessary we use the L-BFGS to optimize the 2D convex dual, which is still fast. For example, for a 17-dimensional action space and a batch size of 512, such as in the Humanoid-v2 experiments, the layer takes roughly 170ms for the forward pass and 3.5ms for the backward pass if the all 512 Gaussians are actually projected[2]. If none of the Gaussians needs to be projected its less than 1ms for forward and backward pass.

**Simplifications** If only diagonal covariances are considered the implementation simplifies significantly, as computationally heavy operations (matrix inversions and cholesky decompositions) simplify to pointwise operations (divisions and square roots). If only the covariance part of the KL is projected, we set $\mu_{\mathrm{old}} = \mu = \tilde{\mu}^*$ and $d\mu = 0$ which is again a simplification for both the derivations and implementation. If an entropy equality constraint, instead of an inequality constraint, it is sufficient to remove the $\omega > 0$ constraint in the dual optimization.

---

[2]On a 8 Core Intel Core i7-9700K CPU @ 3.60GHz

## C  ADDITIONAL RESULTS

Figure 4 shows the training curves for all Mujoco environments with a 95% confidence interval. Besides the projections we also show the performance for PAPI and PPO. In Figure 5 the projections also leverage the Entropy control based on the results from from Akrour et al. (2019).

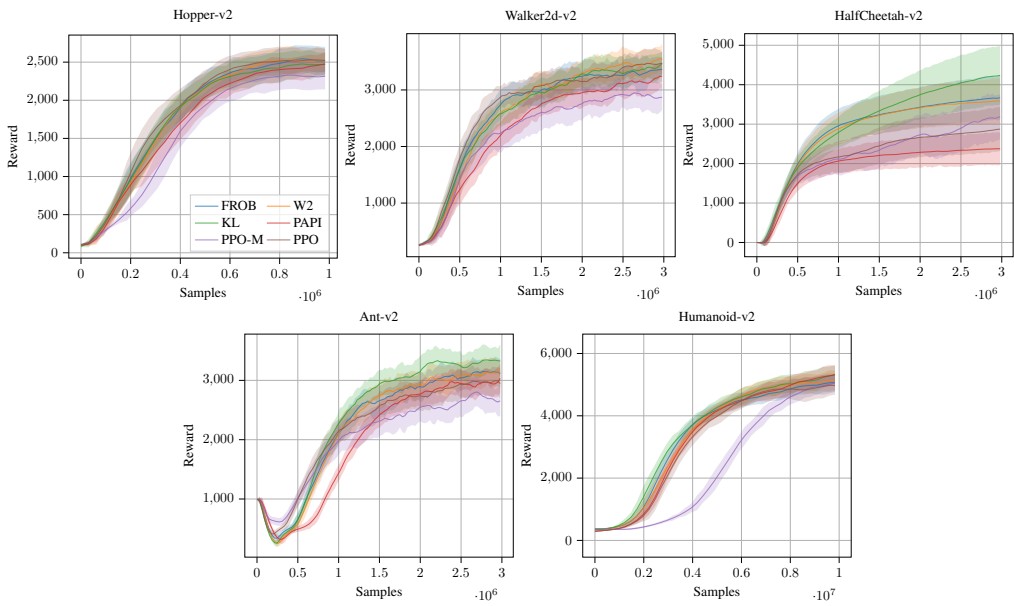

Figure 4: Training curves for the projection layer as well as PPO and PAPI on the test environment. We trained 40 agents with different seeds for each environment using five evaluation episodes for every data point. The plot shows the total mean reward with 95% confidence interval.

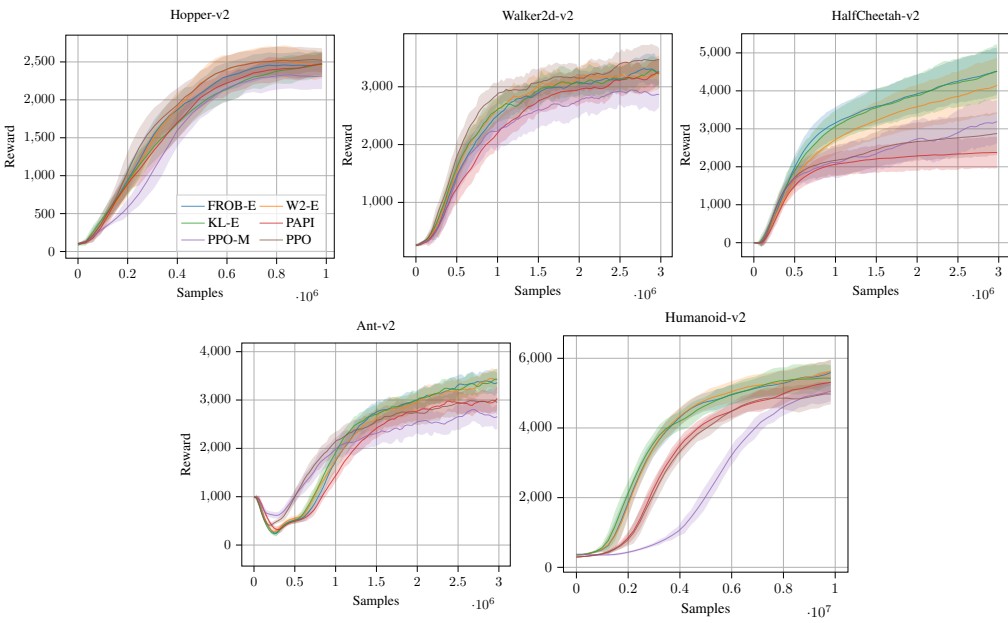

Figure 5: Training curves for the projection layer with entropy control (-E) as well as PPO and PAPI on the test environment. We trained 40 agents with different seeds for each environment using five evaluation episodes for every data point. The plot shows the total mean reward with 95% confidence interval.

# D  HYPERPARAMETERS

Tables 2 and 3 show the hyperparameters used for the experiments in Table 1. Target entropy, temperature, and entropy equality are only required when the entropy projection is included in the layer, otherwise those values are ignored.

Table 2: Hyperparameters for all three projections as well as PAPI, PPO. and PPO-M on the Mujoco benchmarks from Table 1

| | Frobenius | W2 | KL | PAPI | PPO | PPO-M |
|---|---|---|---|---|---|---|
| rollouts | | | | 2048 | | |
| GAE $\lambda$ | | | | 0.95 | | |
| discount factor | | | | 0.99 | | |
| | | | | | | |
| $\epsilon_\mu/\epsilon$ | | 0.03 | | 0.015 | | n.a. |
| $\epsilon_\Sigma$ | | 0.001 | | | n.a. | |
| target entropy | | 0 | | | n.a. | |
| temperature | | 0.5 | | | n.a. | |
| entropy equality | | False | | False | | n.a. |
| | | | | | | |
| optimizer | | | | adam | | |
| epochs vf | | | | 10 | | |
| epochs | | 20 | | | 10 | |
| lr | | 5e-5 | | | 3e-4 | |
| lr vf | | 4.5e-4 | | | 2.5e-4 | |
| minibatch size | | 32 | | | 64 | |
| trust region loss weight $\alpha$ | | 8.0 | | | n.a. | |
| entropy loss penalty | | | | 0 | | |
| | | | | | | |
| normalized observations | | | | True | | |
| normalized rewards | | False | | | True | False |
| observation clip | | n.a. | | | 10 | n.a. |
| reward clip | | n.a. | | | 10 | n.a. |
| vf clip | | n.a. | | | 0.2 | n.a. |
| importance ratio clip | | n.a. | | | 0.2 | |
| | | | | | | |
| contextual std | | | | False | | |
| hidden layers | | | | [64, 64] | | |
| hidden layers vf | | | | [64, 64] | | |
| hidden activation | | | | tanh | | |

Table 3: Hyperparameters for all three projection as well as PAPI and PPO on the Humanoid-v2 from Table 1.

| | Frobenius | W2 | KL | PAPI | PPO | PPO-M |
|---|---|---|---|---|---|---|
| rollouts | | | 16384 | | | |
| GAE $\lambda$ | | | 0.95 | | | |
| discount factor | | | 0.99 | | | |
| | | | | | | |
| $\epsilon_\mu/\epsilon$ | | 0.05 | | 0.015 | | n.a. |
| $\epsilon_\Sigma$ | 1e-4 | 0.01 | 1e-4 | | n.a. | |
| target entropy | | 0 | | | n.a. | |
| temperature | | 0.2 | | | n.a. | |
| entropy equality | | True | | False | | n.a. |
| | | | | | | |
| optimizer | | | adam | | | |
| epochs vf | | | 10 | | | |
| epochs | | | 10 | | | |
| lr | | 1e-4 | | | 1e-4 | |
| lr vf | | 4.5e-4 | | | 1e-4 | |
| minibatch size | | 256 | | | 512 | |
| entropy loss penalty | | | | 0 | | |
| trust region loss weight $\alpha$ | | 8.0 | | | n.a. | |
| | | | | | | |
| normalized observations | | | True | | | |
| normalized rewards | | False | | | True | False |
| observation clip | | n.a. | | | 10 | n.a. |
| reward clip | | n.a. | | | 10 | n.a. |
| vf clip | | n.a. | | | 0.2 | n.a. |
| importance ratio clip | | n.a. | | | 0.2 | |
| | | | | | | |
| contextual std | | | False | | | |
| hidden layers | | | [64, 64] | | | |
| hidden layers vf | | | [64, 64] | | | |
| hidden activation | | | tanh | | | |

Table 4: Hyperparameters for all three projection as well as PAPI and PPO on our ReacherSparse-v0 task from Figure 2. The second value for $\epsilon_\Sigma$ of the KL projection is the bound when using a contextual covariance.

| | Frobenius | W2 | KL | PAPI | PPO | PPO-M |
|---|---|---|---|---|---|---|
| rollouts | | | 16384 | | | |
| GAE $\lambda$ | | | 0.95 | | | |
| discount factor | | | 0.99 | | | |
| | | | | | | |
| $\epsilon_\mu/\epsilon$ | | 0.03 | | 0.03 | | n.a. |
| $\epsilon_\Sigma$ | 5e-5 | 1e-3 | 5e-5/1e-3 | | n.a. | |
| target entropy | | n.a. | | | n.a. | |
| temperature | | n.a. | | | n.a. | |
| entropy equality | | n.a. | | False | | n.a. |
| | | | | | | |
| optimizer | | | adam | | | |
| epochs vf | | | 10 | | | |
| epochs | | | 20 | | | |
| lr | | 3e-4 | | | 1e-4 | |
| lr vf | | 4.5e-4 | | | 1e-4 | |
| minibatch size | | 256 | | | 512 | |
| entropy loss penalty | | | 0 | | | |
| trust region penalty $\alpha$ | | 8.0 | | | n.a. | |
| | | | | | | |
| normalized observations | | | True | | | |
| normalized rewards | | False | | | True | False |
| observation clip | | n.a. | | | 10 | n.a. |
| reward clip | | n.a. | | | 10 | n.a. |
| vf clip | | n.a. | | | 0.2 | n.a. |
| importance ratio clip | | n.a. | | | 0.2 | |
| | | | | | | |
| contextual std | | | False | | | |
| hidden layers | | | [64, 64] | | | |
| hidden layers vf | | | [64, 64] | | | |
| hidden activation | | | tanh | | | |

