# OpenReview forum: "Differentiable Trust Region Layers for Deep Reinforcement Learning"
_ICLR.cc/2021/Conference — ICLR 2021 Poster_

### Official Review · AnonReviewer2 · 2020-10-28
**Good and well-written paper, but experiment require more analysis**

**Rating:** 7
**Confidence:** 3

**Review:**

### Summary
The paper proposes a way to impose trust region restrictions via projections when doing policy optimisation in Reinforcement Learning. The projections have a closed form and enforce a trust region for each state individually. The authors propose three types of projections based on Frobenius, Wasserstein distances and KL divergence. They compare them to the existing methods (PPO, PAPI) and provide some insights about their behaviour.

### Pros
- The paper is coherent and clearly written.
- The paper has a clear motivation and the research question.
- The paper has an extensive and detailed "Related Work" section.
- I find the analysis of the projections and the result of Theorem 1 extremely interesting and insightful. However, I did not check the proof in the Appendix.
- The Appendix has many details useful for further understanding of their approach and reproducing the results.

### Cons
- I find some claims not supported by enough evidence (see Questions).
- I think the experimental section requires more analysis. It's fine not to beat the existing work with 2x better results, but there should be a thorough discussion of what the results mean (see Questions).
- Related work comes before the Background, and sometimes it's quite difficult to understand the details of the prior work and their relation to the paper I am reviewing. "Projections for Trust Regions" subsections could have more details on positioning/comparing the proposed approach to prior work.

### Reasoning behind the score
I enjoyed reading this paper. The story flows coherently and logically. The problem the authors consider is important, and the proposed solution is reasonable theoretically and practically. However, the paper sometimes makes a claim without providing evidence to support it. The paper compares itself with the existing strong methods, however, I find the results section lacks analysis regarding this comparison.

### Questions to the authors
- Your method can impose per-state trust regions. How can you show that this is beneficial? Why is having this beneficial? Can you provide an ablation for this?
- You claim that your method is "more stable and less dependent on code-level optimizations". I don't think you support this claim anywhere in the paper. How can you support that?
- You mention projections in the introduction, but explain it only in the Related Work section. Can you somehow introduce it earlier?
- In the last paragraph of 'Approximate Trust Regions', you mention RL as Inference, EM and Song et al. Can you explain the pros and cons of their approaches? Why is your approach still needed?
- I think, there should be a clear description of what a 'differentiable projection layer' in more details.
- What is the exact RL algorithm you use for optimisation? Can you provide the pseudocode in the appendix? Can you describe what you mean exactly by "successive policy updates"?
- The plots shades overlap and it's really hard to say which method is better and if that's due to randomness or not. I would like to see more discussion on what the numbers tell us. If your method properly imposes trust regions, but the results are comparable to PPO, does it mean that approximate trust regions are okay? You say that "Standard PPO is using a lot of code-level optimizations which are not used by our approach". How can you interpret your results in the light of this? If your method is comparable to PPO without code-level optimisations, what does this tell about your method? Are there any other existing benchmarks which show the superiority of your method? Can you predict some settings, where you method will be significantly better than PPO?
- In the introduction, you say that 'Due to the approximations, they [PPO, TRPO] violate the constraints or fail to find the optimal solution within the trust region'.  However, you also approximate the trust region in Section 4.4. Why is approximating okay for you, but not okay for PPO/TRPO?

### Additional feedback not affecting the score
- You do not include the initial state distribution to the definition of the MDP, without it, the transition function under the expectation in Equation 1 does not make sense ($\mathcal{T}(\cdot | s_{-1} a_{-1})$ for $t=0$. Same about the trajectory distribution.
- typo "covarinces" on page 5
- It took me a while to parse Equations 3 and 4 before I realised that parameters in the minimisation problem and in the constraint differ. Can you give a reference to such an optimisation problem in the existing literature (e.g. Boyd's book or somewhere else)?
- In Section 4, when you describe the projections and Lagrangian multipliers, the results come a bit out of the blue (e.g. 4.1). You have more details in the Appendix, but you do not refer to them from the main text.
- "We trained ten agents on different seeds for each method" in Table 1 caption sounds a bit confusing. Should it be 'for each environment'? Otherwise, it sounds as if you used 10 seeds for a method (2 per each of the five environments).

---

> ### Author Response · Authors · 2020-11-20
> **Author answer (Part 1)**
>
> Thank you for you valuable and appreciated feedback.
> We would like to address your concerns point-by-point in the following.
>
> ### Agnostic to Code-level Improvements
>
> We complemented our studies with an additional baseline, PPO-M, which supports our claim of higher robustness to code-level optimizations.
> PPO-M is leveraging the same minimal set of code-level improvements (for details we refer to the appendix) we used for our projections, however, it is trained with the clipped PPO loss.
> PPO-M, which effectively has the same precondition as our projections, performs worse on all tasks besides on the Hopper-v2 were it achieves comparable performance.
> As Engstrom et al., 2020 already demonstrated, those code-level choices are key to the success of PPO, while our projections do not require them to achieve similar or better performance then the standard PPO.
> Consequently, our projections present a more theoretically and mathematically sound approach to trust region policy optimization.
> In order to present a clearer picture of the performance differences, we now provide a tabular comparison for most experiments and provide the training curves in the appendix.
> In the light of this, all experiments have now been conducted with 40 seeds each instead of the previously used 10 to also improve the statistical significance.
>
> ### Benefit of State-wise Trust Regions and Comparison vs. PPO
>
> To demonstrate the effect of state-wise trust regions we further present a new reaching task that leverages the benefit of contextual covariances and imposes a much harder exploration problem.
> In this setting we can show that our state-wise trust regions are superior and can make better use of contextual information.
> In this environment the need for properly enforced trust regions becomes more apparent compared to the standard Mujoco benchmarks, which can be solve with rather small exploration.
> We found that PPO cannot properly make use of the contextual covariance parametrization and, further, moves from exploration to exploitation too quickly.
> To support our claim even further, we added more analysis regarding the actual change of the policy distribution for each algorithm to the paper.
> Our findings here also align with our previous claim of robustness to code-level choices.
> PPO-M is taking increasingly large steps, while standard PPO can limit the change in the policy distribution with its code level optimizations to some extend.
> Nevertheless, our projections demonstrate a much more consistent change to policy than all baseline methods.
>
> The approximate trust region in combination with code-level choices may be sufficient for standard Mujoco benchmarks regrading performance, but already provides a poor trust region bound in this setting.
> When more exploration is need, however, this approximation is not suitable.
>
>
> ### Approximate Trust Regions and RL as Inference
>
> The RL as Inference framework and our approach are orthogonal.
> No trust-region constrains follow directly from the inference framework and our projection layers could be added to such approaches, as well as, approaches following from any other formulation.
> That said, constraining E- and M-Step in the EM-like algorithms following from the RL as inference framework is common. (V-MPO, Song et al., 2020, MPO, Abdolameki et al. 2018 , Abdolmaleki, et al 2017)
>
> Regarding Song et al., 2020 as well as the "original" MPO, (Abdolameki et al. 2018): Both constrain the policy update (M-Step) using an expected KL constraint which is opposed to our state-wise constraints.
> Such state-wise constraints allow for better control of the policy update.
> Additionally, they rely on an alternating optimization w.r.t. to the policy parameters and Lagrangian multipliers of the trust region constraints to solve the trust region problem.
> Those trust region problems are solved analytically in our case which results in a much easier optimization as it avoids alternating updates.
>
> To summarize, our approach provides a more general and simpler way of enforcing the trust regions constraints used in many approaches from the RL as inference setting and we believe combining our projections with those approaches is a great direction for further research.

---

> ### Author Response · Authors · 2020-11-20
> **Author answer (Part 2)**
>
> ### Differentiable Projection Layer and Successive Policy Updates and RL Algorithm for Optimisation
>
> To give some more details of our algorithm: we use a simple policy gradient of the importance weighted advantage (see Eq. 12). We use Adam to perform this optimization. As the output layer of our policy is the trust region layer, the optimized policy has to stay close to the behavior policy and, therefore, the maximization of the loss function is robust and stable. The policy gradient takes the projection into account as we also differentiate through the output layer to obtain the policy gradient. Hence, the algorithm is a standard policy gradient, but the architecture of the policy is extended by the trust region layer to enforce stable optimizations. We clarified that in the paper and also added a pseudo code description to Appendix A with some additional information about the successive policy update. By "successive policy updates" we mean the iterations of the reinforcement learning algorithm, i.e. a single policy update obtains policy $\pi^i$ from $\pi\_{\textrm{old}} = \pi^{i-1}$.
>
>
> ### Approximations in PPO and our Method
>
> One of the key differences to PPO and TRPO is that our projections do not approximate the computation of the trust region itself.
> By leveraging the trust region projection as final layer in the policy, we always sample from the ``correct'' Gaussian distribution that is within the trust region. Our approximation only takes place after the final mini-batch update to ensure the new behavior policy for the next epoch is satisfying the trust region without the projection layer. This is enforced by the regression loss introduced in Eq. 12. While this approximation is not necessary from a theoretical point of view, it is a practical requirement. Otherwise, we would need to store all previous policies and compute them for one call of the policy $\pi^i$ as the trust region for $\pi^i$ would depend on $\pi^{i-1}$ while the trust region for $\pi^{i-1}$ depends on $\pi^{i-2}$, and so on. PPO or TRPO never compute the optimal solution for the trust region but use approximations to obtain it. In contrast, we compute the optimal solution and use a regression loss such that the output of the network without trust region layer is close to the optimal solution. As mentioned above, we also included a comparison of the policy change for all algorithms (see Fig. 2). While PPO is able to limit the change with its code level optimizations approximately, our projections present a much more consistent change to policy.
>
> ### About Additional Feedback
>
> - "Mention projections in the introduction": We revised section Introduction (2nd paragraph) to mention the projections here.
>
> - We have revised the MDP formulation and the expected return to incorporate an initial state distribution.
>
> - Equations 3 and 4: Both the objective and the constraints are functions of the same optimizing variables. We have elaborated the text below them to explicitly define the optimizing variables.
>
> - We revised the text in Section 5 to correct typos, reference for the results in Section 4, and the description how we use different seeds, e.g. captions in Fig.4 and 5, etc.
>
>
> We hope this alleviates the remaining concerns and again thank the reviewer for their time and feedback.
>
> #### References
> - Abdolmaleki, A., Price, B., Lau, N., Reis, L. P., \& Neumann, G. (2017). Deriving and improving CMA-ES with information geometric trust regions. In Proceedings of the Genetic and Evolutionary Computation Conference (Vol. 8, pp. 657–664).
> - Abdolmaleki, A., Springenberg, T., Tassa, Y., Munos, R., Heess, N., \& Riedmiller, M. (2018). Maximum a Posteriori Policy Optimisation. In International Conference on Learning Representations.
> - Engstrom, L., Ilyas, A., Santurkar, S., Tsipras, D., Janoos, F., Rudolph, L., \& Madry, A. (2020). Implementation Matters in Deep Policy Gradients: A Case Study on PPO and TRPO. In International Conference on Learning Representations.
> - Song, H. F., Abdolmaleki, A., Springenberg, T., Clark, A., Soyer, H., Rae, J. W., Noury, Seb, Ahuja, Arun, Liu, Siqi, Tirumala, Dhruva, Heess, Nicolas, Belov, Dan, Riedmiller, Martin \& Botvinick, M. M. (2020). V-MPO: on-policy maximum-a-posteriori policy optimization for discrete and continuous control. In International Conference on Learning Representations.

---

> > ### Comment · AnonReviewer2 · 2020-11-22
> > **Response to response**
> >
> > Thank you! I think the authors did a good job of improving the paper and addressing my concerns. I will consider updating the score after a discussion with the other reviewers/AC.

---

> > > ### Author Response · Authors · 2020-11-24
> > > **Thank you for the update**
> > >
> > > Thank you again for reviewing our revised work. We appreciate the update and are glad we could address your concerns.

---

### Official Review · AnonReviewer3 · 2020-10-29
**Review for Paper3480**

**Rating:** 6
**Confidence:** 5

**Review:**

### Summary
In trust-region-based policy optimization methods such as TRPO and PPO, it is difficult to tune and lots of approximations are required. The authors try to solve this issue by introducing the closed-form derivation of trust regions for Gaussian policies with three different types of divergence (or distance). Based on the theoretical derivation, the differentiable layer is proposed, where the layer is built upon “old” policy during the trust-region-based policy updates. The difference comes from the use of various divergences (or distances) are given in theoretical and empirical ways.

### Quality
The proposed idea seems interesting and theoretical derivations seem mostly sound but empirical performance doesn’t support the authors’ claim, which makes me highly decrease the score.

### Clarity
- The Introduction is written mostly well although some sentences are too specific to be understandable at the first glance.
- The Related Work is well-summarized and emphasizes the difference and advantages clearly.
- Section 4 (which is about the main ideas) needs to be more clarified and reorganized for some parts.
- The use of entropy projection is quite abrupt and I couldn’t understand its clear motivation.
- Successive policy updates seem to improve the methods in a way that the mean is well-bounded during updates, but its explanation and justification are difficult to understand.
- There are more comments in `Detalied Comments`

### Originality
- I really enjoyed reading projections of mean and covariance (~4.2) and ideas are somewhat novel. However, empirical results don’t support the authors’ claim in the sense that the differentiable trust-region layer doesn’t improve the performance significantly.

### Significance
- The idea is interesting, but its significance is low due to the empirical performances as well as loosely tuned baseline (PPO).
- Additionally, some motivations/derivations/links among equations are unclear.

### Detailed comments
(p.1, Abstract) code-level optimization
- I was curious about the definition at the first glance.

(p.1, Introduction) our method comes with the benefit of imposing the constraints on
the level of individual states, allowing for the possibility of context dependent trust regions.
- I couldn’t understand what this means at the first glance.

(p.1, Introduction) Considering the trust region layers require
- Considering the trust region layers requires

(p.1, Introduction) the new policy without trust layers has to stay close to the output of the projection layer.
- I don’t think such a specific methodology needs to appear in the introduction.

(p.2, Related Work) However, Engstrom et al. (2020) and Andrychowicz et al. (2020) recently showed that code-level optimizations are essential for achieving state-of-the-art results with PPO
- The meaning of code-level optimization needs to be clearer.

(p.3, Related Work) They use either an exponential or linear decay of the entropy during policy optimization to control the exploration process and escape local optima. To leverage those benefits, we embed this entropy control mechanism in our differentiable trust region layers.
- I’d rather add simple maths to describe what the previous works did for clarity.

(p. 3, Preliminaries and Problem Statement) Eq. (1)
- In the subscript of the expectation, I’d rather add either $t=1, …$ or trajectory distribution.
- $A^\pi$ -> $A^{\pi_{\mathrm{old}}}$?
- I think using bold letters for all state-action pairs is a bit confusing. I’d use bold letter only for random variables and plain letters for non-random variables.

(p.3, Preliminaries and Problem Statement) Using a constraint on the maximum KL over the states has been shown to guarantee monotonic improvement of the policy (Schulman et al., 2015a). However, as all current approaches do not use a maximum KL constraint but an expected KL constraint, thus the monotonic improvement guarantee also does not hold exactly.
- I think this sentence should be emphasized.

(p.3, Preliminaries and Problem Statement) Gaussian policies ~ as well as
- as well as -> and?

(p.3, Preliminaries and Problem Statement) the commonly used reverse KL
- The expression `commonly used` here seems weird to me.

(p.3, Preliminaries and Problem Statement) The similarity of the covariance is measured by the difference in entropy of both distributions
- This statement seems incorrect since entropy is an averaged valued of negative log probability and KL is not exactly the difference between two entropies.

(p.3, Preliminaries and Problem Statement) as it is always non-negative
- since it satisfies the criteria of being a divergence.

(p.3, Preliminaries and Problem Statement) as the distance measure is then more sensitive to the data-generating distribution.
- If I understood correctly, the distance is defined by using the covariance of the old policy distribution (similar to the distance proposed in Dadashi et al., 2020 that only cares about diagonal covariance matrix), but how is this related to the sensitivity w.r.t. the data-generating distribution?

(p.4, Differentiable Trust-Region Layers for Gaussian Policies) Additionally, we extend the trust region layers to include an entropy constraint to gain control over the evolution of the policy entropy during optimization
- I’d rather use this sentence where the formula for entropy constraint appears.

(p.4, Differentiable Trust-Region Layers for Gaussian Policies) The trust regions are defined by means of a distance or divergence ~
- In my understanding, it’s not a mean over distance, but just a distance between new and old policies.

(p.4, Differentiable Trust-Region Layers for Gaussian Policies) Note that $\mu$, $\Sigma$ are state-dependent, which we will however neglect for ease of notation.
- I’d rather keep the dependencies on states since it’s a bit confusing to understand the state dependencies of objectives (3) and (4).
- If I understood correctly, (3) and (4) will be optimized for each $s\in\mathcal{S}$, and thus, we can use the solution of projection over all states the old policy can visit.

(p.4, Differentiable Trust-Region Layers for Gaussian Policies) The output of the trust region layer is then considered to be the new policy.
- Since (3) and (4) are objectives, I’d rather state like “We desire the output of the trust region layer becomes the parameters of the new policy and formulate the trust region layer as follows.”

(p.4, Differentiable Trust-Region Layers for Gaussian Policies) As all distances or divergences used in this paper can be decomposed into a mean and a covariance dependent part
- This may be since we don’t update the second distribution of all metrics -- therefore, $\Sigma_2$ is fixed -- where the old policies will be plugged in. Such an explanation seems to be needed.

(p.4, Differentiable Trust-Region Layers for Gaussian Policies) as this gives the algorithm more flexibility
- I understand the way the trust region will be used, but this statement is weird since joint optimization is much more general and flexible in my understanding.

(p.4, Differentiable Trust-Region Layers for Gaussian Policies) where $d_\mu$ is the mean dependent part and $d_\Sigma$ is the covariance dependent part of the employed similarity measure
- The subscripts are confusing since $\mu$ and $\Sigma$ is used as input arguments of (3) and (4). $\mu$ and $\Sigma$ are the output parameters of Gaussian policy which seem to be fixed during the optimization of (3) and (4). This should be stated for clarity.

(p.4, Differentiable Trust-Region Layers for Gaussian Policies) All three trust region projections
- “Three” indicates different distances, so it should be linked with the previous equations on distances.

(p.4, Differentiable Trust-Region Layers for Gaussian Policies) By making use of the method of Lagrangian multipliers,~
- Link Appendix A.2. for readers.

(p.5, Wasserstein Projection) To find the projected covariance ~
- I’d rather put this sentence after the sentence “However, in practice we found this approach to be numerically unstable.”

(p.5, Wasserstein Projection)  For the more general case of arbitrary covariance matrices, we would need to ensure the matrices are sufficiently close together, which is effectively ensured by Equation 6.
- I don’t fully understand what the authors intended to say.

(p.5, Wasserstein Projection)  Note however, that here we chose the square root of the covariance matrix ~
- Is there an advantage of using the square root of the covariance matrix? Also, it would be helpful if the definition of the square root of the matrix is given.

(p.6, Figure 1) Entropy of the interpolated covariances
- Covariance cannot define entropies. We should use “Entropy of the interpolated distributions”

(p.6, Entropy Projection)
- This is a bit abrupt to me since trust region w.r.t. old policy has been considered until page 5. At the first glance, I couldn’t understand why entropy projection is needed and how the scaling is related to the exploration. A more intuitive explanation is needed.

(p.6, Analysis of the Projections) A paragraph with the sentence “It is instructive to compare the three projections.”
- I’d rather use equations for a detailed explanation.

(p.7, Successive Policy Updates) The above projections can directly be implemented for training the current policy. However, for successive policy updates the projections become rather impractical. Each projection would rely on calling the projection of the preceding policy.
- I couldn’t understand this part. My best understanding was using stated projections for policy updates is impractical, but how each projection is related to preceding policy is unclear.



(p.7, Successive Policy Updates) The most intuitive way to solve this problem is to use the existing samples for additional regression steps after the policy optimization. Yet, this adds a computational overhead.
- I couldn’t understand this part.

(p.7, Successive Policy Updates) Eq (10)
- $\tilde{\pi}$ seems ”Differentiable trust-region layer” and needs to be mentioned.

(p.7, Experiments) For our experiments, the PPO hyperparameters are based on the original publication (Schulman et al., 2017). The PAPI projection as well as its conservative PPO version are executed in the setting sent to us by the author. For our projections, all parameters are selected with Optuna (Akiba et al., 2019)
- PPO is a baseline here but seems naively tuned.

(p.7, Table 1) We trained ten agents on different seeds
- It seems ten seeds, not ten agents?
- PPO works much better in Hopper and Walker2d, which is different from the statement in the main context (“The results show that our trust region layers are able to perform similar or better than PPO and PAPI across all tasks”)

(p.7, Figure 2) Left
- The result doesn’t seem statistically significant. Means of proposed methods are within the confidence interval of PPO.

### References
- Dadashi et al., 2020, “Primal Wasserstein Imitation Learning”

---
### Response to Authors
I'm satisfied with your responses, especially strengthened experimental results and the clarification of methods in the revision. As my concerns were on doubtful empirical results in the submission---although I thought the approach of the submission was sufficiently novel---I updated my score from 4 to 6 accordingly. I don't know if the authors will keep working on this direction, but I think it will be interesting to see the performance of *off-policy RL* with the proposed method.

---

> ### Author Response · Authors · 2020-11-20
> **Author answer (Part 1)**
>
> Thank you for your valuable feedback and the very detailed remarks. We incorporated the suggestions and hope this improves clarity. Additionally, we now included an algorithmic view of our approach in the Appendix A to clarify the actual algorithm used in the experiments. Aside from the minor corrections, we want to address in the following your specific question in more detail.
>
> ### Performance and Significance of our Contribution
>
> As there are some concerns from the reviewer regarding the tuning of PPO and the significance of our experimental results, we conducted additional experiments. We now use 40 seeds instead of 10 for all experiments in our paper. The new results show that we are able to perform better on all tasks, besides the Hopper-v2, where we achieve a comparable performance. However, we have to agree the differences are small and we do not massively outperform PPO on the Mujoco benchmarks. In the light of this, we want to politely point the reviewer to the Reviewer Guidelines stating that not clearly performing state-of-the-art is on its own not a reason for rejection (https://iclr.cc/Conferences/2021/ReviewerGuide\\#faq). While we do achieve state-of-the-art, we provide mathematically more principled trust region layers that are agnostic with respect to the reinforcement learning algorithm (e.g. they could also be used for actor critic methods) and do not rely on code-level optimizations. To support the claim of the latter, we added an additional baseline, PPO-M, that uses the PPO objective, but the same minimal implementation choices as we do. We show that in this setting PPO performs clearly worse than our projections. Additionally, we complemented the paper with an analysis of the actual change in the policy distribution. Unlike PPO and PAPI, the policy generated by our projections changes much more consistently and, in particular, PPO-M is taking increasingly large steps.
>
> To emphasize the benefit of state-wise trust regions, we included one additional experiment for a reaching task that involves a much harder exploration problem than the standard Mujoco tasks. Here, we can clearly outperform PPO with and without code-level improvements. Moreover, in combination with state-dependent covariances our trust-regions benefit even more. PPO, on the other hand, cannot make use of it and its performance deteriorates. This example shows that, due to the state-individual trust regions, our approach comes with the promise to scaling to more complex policy structures and exploration problems. Yet, this advantage is not visible for the standard Mujoco benchmarks as the exploration problem in these examples seem to be too simple to exploit state-dependent covariances.
>
> Hence, we believe that, as we achieve comparable performance than state-of-the-art and have beneficial properties, that the paper qualifies for acceptance even though we do not massively outperform PPO.
>
> ### Hyperparameter Tuning of PPO
>
> To also address the tuning of PPO, the PPO hyperparameters for the four basic Mujoco tasks have now also been tuned with Optuna (Akiba et. al., 2019).
> However, the best set of parameters was on par with our previously used default, as a consequence, we left the parameters unchanged.
> For both the Humanoid-v2 as well as for our newly added Reacher experiment we found better performances by using Optuna.
> We think our initial parametrization for the other four tasks was already a sufficiently good benchmark of PPO's performance.
> In our experiments PPO performs equally well to the Spinning-Up implementation benchmarks (https://spinningup.openai.com/en/latest/spinningup/bench.html) and e.g. the performance in Fujimoto, et al., 2018.
> Further, the tuned parameters for the stable-baseline zoo (https://github.com/araffin/rl-baselines-zoo) are equivalent to ours and even used as a default for the new *stable-baselines3*.
> Even other popular implementations such as https://github.com/ikostrikov/pytorch-a2c-ppo-acktr-gail leverage the same parameters as default.
>
> ### Code-level optimizations
>
> When we speak of code-level optimizations we always refer to choices that are described by Engstrom, et al., 2020, but we agree that we needed to highlight this fact more.
> Regarding the mathematical description of an exponential or linear decay in the related work section.
> We did refrain from adding this as we find that the related work part should emphasize the connection to other work and, therefore, avoid  introducing mathematical notation if it is not essential for doing so.
> For more details on that we therefore refer to the algorithmic view of our approach in the Appendix A where we demonstrate how to apply the entropy control.

---

> ### Author Response · Authors · 2020-11-20
> **Author answer (Part 2)**
>
> ### Maximum KL constraint and Asymptotic Convergence
>
> We agree that this point should be stressed more. Therefore we now mention this important difference of our approach in comparison to previous works already in the introduction and again in the conclusion.
>
> ### Trust Region Measures
>
> We improved our description for the reverse KL divergence.
> The covariance part is indeed given by two components, the difference in scale by the log ratio of the determinants (which relates to the difference in entropy of both distributions) as well as the rotation difference by the trace term.
> Both is mentioned explicitly in the revision of the paper.
>
> The metric space used for the Wasserstein distance relates to the sensitivity of the data-generating distribution, because the data is generated by the old distribution, i.e. sampled from a Gaussian distribution whose covariance matrix is given by $\Sigma\_{\textrm{old}}$. Thus using $\Sigma\_{\textrm{old}}$ to measure the distance between the old and the new distributions explicitly takes the generating distribution of the data into account.
>
> ### Theoretical Considerations of Differentiable Trust-Region Layers
>
> We agree that the dependence of Eq. 3 and 4 on the state might not be obvious, therefore, we keep the state dependence in these equations and only omit it afterwards. This should help the reader to get a better understanding of the dependence of the constraints and parameters on the state.
>
> Considering the formulation of the problem of Eq. 3 and 4. While joint optimization is more general, the additional flexibility in treating $\mu$ and $\Sigma$ separately comes from the fact that we can introduce different constraints for the two parameters, thus allowing e.g. $\Sigma$ to deviate more from its previous value than $\mu$ and vice versa. This yields better results and is also common practice in black-box optimization (Abdolmaleki, et al., 2017).
>
> We now go into more detail about the commutativity assumption for the general W2 projection as well as the corresponding requirement that the matrices have to be sufficiently close together. First, the simplified expression for the solution only holds for the case of commuting covariance matrices. For the special case of diagonal covariances, the commutativity always holds. In the general case, however, the simplified expression becomes an approximation, which is justified only when the covariance matrices are sufficiently similar, such that the error made by assuming commutativity is small. This assumption, i.e. the similarity of the two covariance matrices, is required by the bound on the covariance matrix in Eq. 8.
>
> Lastly, the square root of the covariance matrix appears naturally in the expression for the solution of the projected covariance matrix (Eq. 9). Thus, parametrizing the algorithm in terms of the square root of the covariance matrix instead of its Cholesky factors leads to simpler computations and increased numerical stability. We extended the corresponding footnote, please have a look there.
>
> ### Entropy Control
>
> For the Gaussian policies considered in this work the entropy directly relates to the "size" of the covariance, which again relates to the amount of exploration in the on-policy setting in our work. Previous works (Abdolmaleki et al., 2015, Pajarinen et al., 2019, Akrour et al., 2019) have demonstrated that controlling the entropy, and, thus, the amount of exploration can yield to improved performance, especially when combined with trust regions. We decided to include it in our work as we saw similar effects for our approach and it can be easily incorporated in the trust region layers. We, however, tried to make this fact more apparent in our paper.
>
> ### Successive Policy Updates
>
> We added pseudo code to Appendix A with some additional information about the successive policy update and, moreover, improved the corresponding section 4.4.
> To give some more details here: The successive policy update is effectively the update of the behavior/old policy $\pi\_{\text{old}}$ that is used for generating trajectories.
> When using other on-policy methods, such as PPO, we naturally obtain this policy by choosing the most recent set of parameters for each epoch $i$, i.e. $\pi\_{\text{old}}^{i} = \pi^{i-1}$.
> At each epoch $i$, however, the policy $\pi^i$ predicted by the network does not respect the constraints before the projection layer, thus it relies on calling this layer. Yet, the policy of the projection layer $\tilde{\pi}$ depends on the parameters of $\pi^i$ and the old policy network $\pi^i\_\textrm{old} = \tilde{\pi}^{i-1}$. This would result in an ever-growing stack of policy networks which would become increasingly costly to evaluate. In other words, $\tilde{\pi}^i$ is computed using all stored networks of $\pi^i,\pi^{i-1}, \ldots,\pi^0$. As a consequence, we need to encode the information of the projection layer into the parameters of $\pi^i$, which is done by the regression penalty in Eq. 12.

---

> ### Author Response · Authors · 2020-11-20
> **Author answer (Part 3)**
>
>
> ### Remaining Corrections:
>
> - We fixed the problem statement in Eq. 1 according to the reviewer's comment.
> - We clarified the description of the KL as a divergence.
> - We improved clarity and the explanation of the general projection problem in Eq. 3 and 4, especially with regard to the individual components involved.
> - We added references to the detailed derivations in the appendix.
> - We corrected all unsuitable statements, typos, and grammatical errors are corrected; and elaborated at confusing sentences according to the reviewer's detailed comments.
>
> We hope this alleviates the remaining concerns and again thank the reviewer for their time and detailed feedback.
>
> #### References
>
> - Abdolmaleki, A., Lioutikov, R., Peters, J. R., Lau, N., Reis, L. P., \& Neumann, G. (2015). Model-Based Relative Entropy Stochastic Search. In Neural Information Processing Systems (pp. 3537–3545).
> - Abdolmaleki, A., Price, B., Lau, N., Reis, L. P., \& Neumann, G. (2017). Deriving and improving CMA-ES with information geometric trust regions. In Proceedings of the Genetic and Evolutionary Computation Conference (Vol. 8, pp. 657–664).
> - Akiba, T., Sano, S., Yanase, T., Ohta, T., \& Koyama, M. (2019). Optuna: A Next-generation Hyperparameter Optimization Framework. Proceedings of the ACM SIGKDD International Conference on Knowledge Discovery and Data Mining, 2623–2631.
> - Akrour, R., Pajarinen, J., Neumann, G., \& Peters, J. (2019). Projections for Approximate Policy Iteration Algorithms. In Proceedings of the 36th International Conference on Machine Learning (pp. 181–190).
> - Engstrom, L., Ilyas, A., Santurkar, S., Tsipras, D., Janoos, F., Rudolph, L., \& Madry, A. (2020). Implementation Matters in Deep Policy Gradients: A Case Study on PPO and TRPO. In International Conference on Learning Representations, ICLR.
> - Fujimoto, S., Van Hoof, H., \& Meger, D. (2018). Addressing Function Approximation Error in Actor-Critic Methods. In 35th International Conference on Machine Learning, ICML (Vol. 4, pp. 2587–2601).
> - Pajarinen, J., Thai, H. L., Akrour, R., Peters, J., \& Neumann, G. (2019). Compatible Natural Gradient Policy Search. Machine Learning, 108(8–9), 1443–1466.

---

### Official Review · AnonReviewer4 · 2020-11-02
**Differentiable Trust Region Layers for DRL - why these three?**

**Rating:** 6
**Confidence:** 3

**Review:**

The authors explore the use of KL, 2-Wasserstein, and Frobenius norm in order to derive trust region projections in DRL. The topic is relevant and novel - specially given the prevalence of TRPO and PPO in RL in recent years.

The paper is well-written and structured, with a nicely written related work section. I find the study very relevant and useful but somewhat incomplete. I would like the paper to better motivate

- why these three (KL, Wasserstein, Frobenius) were chosen. By the way, be precise with the use of the word metric (beginning of Sec 4) when referring to KL. It would be nice to extend the same analysis for the wider families of metrics and divergences that these three are part of.

- the rationale to select the right one for a given problem.

I also find that the discussion on entropy control, although interesting on its own, somehow distracts from the main message of the paper.

---

> ### Author Response · Authors · 2020-11-20
> **Author answer**
>
> Thank you for you valuable and appreciated feedback.
> We would like to address your concerns point-by-point in the following.
>
> ### Why KL, Wasserstein, and Frobenius?
>
> We agree that considering different metrics and divergences would be very interesting and instructive.
> Yet, the main problem with using arbitrary metrics and divergences is that the optimization problems resulting from such trust region formulations have, in general, no closed form solution. This prevents any practical realization of those approaches, which scales to problems of relevant size. The three projections we picked are special in the sense that efficient solutions exist. As shown, for the Wasserstein and Frobenius metrics closed form solutions can be derived. For the KL divergence, solving the optimization completely in closed form is impossible, but we can still obtain closed form solutions for the primal and only the dual needs to be solved numerically. As we have a fixed amount of constraints, i.e. $1$, this optimization is much easier than optimizing the primal, whose dimension scales linearly with the action dimension. This dual optimization allowed us to still implement it in efficient manner that scales to problems of relevant complexity.
>
>
> ### Preferred Choice of Projection
>
> To provide some more insight for which scenarios each projection is preferred, we added some more analysis to the initial results and, further, provided an additional experiment with contextual covariances. Generally speaking, the Frobenius projection is the weakest out of all three projections. It tends to run into numerical problems when covariances are small at the end of the training, especially with contextual covariances. Tighter covariance bounds and higher weights for the regression penalty in the loss can, however, mitigate those effects. The KL performs well overall, similar to the W2, hence, if contextual covariances are not required, the KL is the best choice for most problems. As a bonus, it has all properties of existing KL-based trust region methods that have monotonic improvement guarantees. Nevertheless, for quick benchmarks with contextual covariances, the W2 is preferred, given it does not add any computational overhead as the KL does. More specifically, the contextual covariance for the KL requires about ten times more compute time than the Frobenius or W2 projection. This, however, is only for tight covariance bounds and we currently work to reduced that by improving the initialization of the dual variables.
>
>
> ### Entropy Control
>
> Although the entropy control mechanism discussed is not itself a contribution of this work, previous works have shown that entropy control can improve performance of RL algorithms, especially when combined with trust regions (Abdolmaleki et al., 2015, Pajarinen et al., 2019. Akrour et al., 2019). We decided to include it in our work as we saw similar effects for our approach and it can be easily incorporated in the trust region layers. We also believe that improved schemes of controlling the trust regions and entropies are a promising direction for further research. Yet, to address your concerns, we reformulated the corresponding parts to reduce the emphasis on the entropy control, especially in Section 4.
>
> ### Remaining Corrections:
>
> - We made sure to be more precise with the terms metric and divergence throughout the whole work.
>
> We hope this alleviates the remaining concerns and again thank the reviewer for their time and feedback.
>
>
> #### References
>
> - Abdolmaleki, A., Lioutikov, R., Peters, J. R., Lau, N., Reis, L. P., \& Neumann, G. (2015). Model-Based Relative Entropy Stochastic Search. In Neural Informaion Processing Systems (pp. 3537–3545).
> - Akrour, R., Pajarinen, J., Neumann, G., \& Peters, J. (2019). Projections for Approximate Policy Iteration Algorithms. In Proceedings of the 36th International Conference on Machine Learning (pp. 181–190).
> - Pajarinen, J., Thai, H. L., Akrour, R., Peters, J., \& Neumann, G. (2019). Compatible Natural Gradient Policy Search. Machine Learning, 108(8–9), 1443–1466.

---

### Official Review · AnonReviewer1 · 2020-12-04
**Belated Review (Not Considered as an Official Review in the Final Decision)**

**Rating:** 6
**Confidence:** 4

**Review:**

I’m terribly sorry but I noticed that somehow my review of this paper was not successfully submitted as I checked my submission tasks. After double checking with the area chair, we decide to add my review here. Notice that this is **only for the authors' reference and to provide some additional feedback for potential improvement of the paper in the future**, but it is **not considered as an official review in the final decision process**. Please find both the original review and the comments on the revised paper below.

### [Original Review]

This paper considers trust region methods in deep reinforcement learning (DRL) and proposes differentiable trust region layers, a type of differentiable neural network layers to enforce state-wise trust regions exactly for deep Gaussian policies via closed-form projections. The proposed approach is flexible and general, and can in particular handle different trust regions (in KL, W2 and Frobenius norms, for example) and can be applied to existing RL algorithms as a complementary component. Empirical results show that the proposed trust region layers (together with entropy projection to encourage exploration) help PPO achieve similar or better results, and are less dependent on implementation/code-level optimization.

In general, the paper is relatively well-written and discusses about a novel and clean approach for solving the problem of enforcing trust region constraints in trust region DRL algorithms. However, the following issues should be noted and addressed:
1. On page 2, the authors mention that “Additionally, the projection is not directly part of the policy optimization but applied afterwards, which can result in suboptimal policies”. But since the subproblems of TRPO/PPO are just (first-order) approximations to the original RL problem, the meaning of “suboptimal” here is unclear.
2. On a related point, I’m wondering what would happen if the authors instead use the projection methods in this paper not as a layer but just as a post-processing step after each standard TRPO/PPO update. The authors should compare this approach with the proposed one (Section 4.4), as this is at least a natural and closely related benchmark (and may even perform better in practice, which is currently unclear without the numerical comparisons). Also, the authors mention that “for successive policy updates the projections become rather impractical”. However, it is not clear to me why it is impractical to use the projection as a post-processing step as mentioned above.
3. Again, on a related point, in Section 4.4, it is unclear which policy is eventually adopted in execution. Is it the $\pi_{\theta}$ (before projection) or the projected policy $\tilde{\pi}$?
4. Why is it that important to enforce the trust regions exactly? It seems that the major reason provided in this work for focusing on this problem is that exact trust region constraints will lead to some performance improvement with less code-level optimization. However, in general, the empirical performance improvement shown in this paper is not very significant. In fact, for Table 1, I don’t think the two criteria (“first” and “last”) are informative enough to characterize the overall performance, as it seems that the curves are crossing each other very frequently (in Figures 2 and 4), and so it would make a lot of difference to consider the last 20 epochs, 10 epochs, or just 5 epochs and so on. So it might be better to directly look at the curves. However, from Figures 2 and 4, it seems that with only the trust region layers, the performance improvement is not very obvious. It is only with the additional entropy projection that the performance becomes obviously better (Figure 2). Hence I think the authors should also include comparisons with the standard PPO/TRPO + entropy projection. Otherwise, it is not clear whether the entropy projection is central or the trust region layers proposed in this paper are central.
5. On a related point, can the authors provide any reasoning about why it is important to enforce the trust region constraints exactly from a theoretical viewpoint?
6. Why is it important to avoid code-level optimization? If I understand correctly, code-level optimization are just some tricks commonly adopted in TRPO/PPO methods, as pointed out in (Engstrom et al., 2020). Then why is it a big issue to need code-level optimization?

There are also some slightly more minor issues:
1. In the abstract, the authors mention that existing trust region DRL methods “lack sufficient exploration”. However, as pointed out later in the paper, existing trust region DRL methods like PAPI have proposed to use entropy projection to encourage exploration, and so this claim is not very accurate.
2. Again in the abstract, the authors claim that the proposed differentiable trust region layers can complement existing RL algorithms. However, it seems that the authors only applied these layers to the PPO algorithm. Can the proposed layers also be applied to other RL algorithms (beyond PPO/TRPO)?
3. On page 3, in the definition of the Gaussian policies, the authors may want to make it clearer that $\mu$ and  $\Sigma$ are parametrized by $\theta$ (if it’s the case). Otherwise, it may appear that $\theta$ is simply the concatenation of $\mu$ and $\Sigma$, which would rule out the deep neural network parametrization.
4. On page 4, at the end of Section 3, it would be better to explain why only the metric for $\mu$ is scaled by $\Sigma_2^{-1}$, while the metric for $\Sigma$ is not. It may also be helpful to consider the alternative Frobenius norm with the second term replaced by ${\rm tr}(\Sigma_2-\Sigma_1)^T\Sigma_2^{-1}(\Sigma_2-\Sigma_1)$ and numerically test and compare the performance.
5. What is the “entropy projection on its own” approach? Is it just adding entropy projection on top of (1) without trust region constraints?
6. There seem to be some inconsistencies between the tables and the plots. In particular, for Humanoid-v2, Table 1 shows that KL performs the best in terms of the “last” criteria, but from the center plot of Figure 2, KL seems to be one of the worst in the last epochs. The authors should double check to make sure that there are no such kind of inconsistencies.

### [Comments on the Revised Paper, Rating and Confidence]

The revised version now contains a much clearer description of how the layers are integrated into the algorithm, fixes several typos and reorganizes (and enriches some details of) the numerical experiments following comments of the other reviewers.

However, most of my major concerns above still remain (which is expected as the authors didn’t get a chance to see my review, and I sincerely apologize for this).
1. For example, although the authors now provide some more detailed explanations about “impractical projections” in Section 4.4, it is still unclear why one cannot use the projection as a post-processing step instead of a layer, and what the authors are trying to convey in the more detailed discussion about impractical projections with a growing storage of previous policy networks in the revised draft here.
2. Also, the authors may want to clarify some new terminologies and notation introduced in the revision. For example, are “contextual policies” just policies with state-dependent covariances? And what is the index $t$ in the Adam updates in Algorithm 2, and should $a$ and $s$ also be $a_t$ and $s_t$ here?
3. Another issue I noticed is that compared with the revised draft, the results (in terms of which method is optimal, and whether or not the proposed layers improve over PPO/PAPI) shown in the plots and the tables are not very stable, which indicate that different runs give pretty different results. Such kind of instability may also be relevant to the inconsistency between Table 1 and Figure 2 in the original draft mentioned in the original review above.

Overall, I decide to maintain my original rating.

---

### Decision · Program_Chairs · 2021-01-07
**Final Decision**

**Decision:**

Accept (Poster)

**Comment:**

This paper proposes a differentiable trust region based on closed-form projects for deep reinforcement learning. The update is derived for three types of trust regions: KL divergence, Wasserstein L2 distance, and Frobenius norm, applied to PPO and PAPI, and shown to perform comparably to the original algorithms.

While empirically the proposed solutions does not bring clear benefits in terms of performance, as correctly acknowledged by the authors, it is rigorously derived and carefully described, bringing valuable insights and new tools to the deep RL toolbox. The authors improved the initial submission substantially based on the reviews during the discussion period, and the reviewers generally agree that the work is of sufficient quality that merits publication. To improve the paper and its impact, I would recommend applying the method to also off-policy algorithms for completeness. Overall, I recommend accepting this submission.